# VIBE: Disentangling Social Dynamics via Kinematics-Informed Variational Inference for Behavioral Emotion

Abhishek Pratap Singh [1]   Vaibhav Pratap Singh [2]   Deepak Kumar [1]   Balasubramanian Raman [1]

## Abstract

Group Emotion Recognition (GER) is crucial for understanding social dynamics, ranging from interpreting intimate conversations to evaluating crowd behavior in large-scale surveillance scenarios. While current AI models can analyze these scenes, they often act as *black boxes* that take shortcuts. Instead of focusing on how people are actually behaving, these models often get distracted by the background environment, leading to inaccurate results. To bridge this gap, we introduce **VIBE** (**V**ariational **I**nference for **B**ehavioral **E**motion), a kinematics-aware framework that integrates audio, video, and text modalities through causal structuring. Unlike standard models that simply mix data together, VIBE utilizes mathematical constraints to filter out background noise and isolate the genuine emotions of the people involved. This purified representation enables our model to focus exclusively on the sociological mechanics of the crowd, dynamically modulating neural attention based on raw physical synchrony. Simultaneously, we align visual dynamics with human interpretability by projecting latent representations into a semantically structured space informed by textual descriptions. Comprehensive experiments demonstrate that VIBE consistently outperforms state-of-the-art methods. Code is available at GitHub.

## 1. Introduction

Human emotion is more than just a facial expression; it is a shared experience that grows and changes as we interact with the people around us. While Affective Computing has matured significantly in analyzing individuals, GER remains a frontier challenge with critical applications in social robotics (Huang et al., 2025; Abbas et al., 2025), large-scale event surveillance, and crowd analysis.

The field has recently shifted from controlled laboratory settings to the chaotic reality of user-generated content, driven by "in-the-wild" datasets such as VGAF (Sharma et al., 2019; 2021) and GECV (Quach et al., 2022). Consequently, the dominant paradigm has evolved from hand-crafted features to end-to-end deep learning. State-of-the-art approaches, including Hybrid Networks (Pinto et al., 2020), Cross-Modal Attention mechanisms (Wang et al., 2020; Praveen & Alam, 2024), and Transformer-based fusion architectures (Waligora et al., 2024), have demonstrated impressive capabilities in mapping raw pixel and acoustic data to categorical emotional labels.

Despite these performance gains, current methodologies face a critical theoretical bottleneck: they operate as *discriminative black boxes*. Existing frameworks typically rely on massive, pre-trained backbones (e.g., ResNet (He et al., 2016), ViT (Dosovitskiy et al., 2021), HuBERT (Hsu et al., 2021)) to extract high-dimensional features, which are then fused via concatenation or standard attention pooling (Quach et al., 2022; Kumar et al., 2024b). Although effective at surface-level feature recognition, this approach neglects the foundational sociological and physical principles that drive collective behavior (Sun et al., 2024).

Specifically, current models fail to account for three critical dimensions:

- **Entanglement:** Standard encoders inevitably entangle robust emotional signals with nuisance factors (e.g., camera motion, lighting conditions) (Arjovsky et al., 2020). Without causal intervention, models often learn spurious correlations, leading to poor generalization on unseen environments.

- **Physical Synchrony:** Psychological theory posits that "Group Cohesion" the synchronization of physical movement, is a primary indicator of group valence and arousal (Wlodarczyk et al., 2020). Pure deep learning models must relearn these kinematics from scratch, often inefficiently, rather than having them enforced at

---

[1]Department of Computer Science and Engineering, Indian Institute of Technology Roorkee, Roorkee, India [2]Department of Computer Science and Engineering, Malaviya National Institute of Technology Jaipur, Jaipur, India. Correspondence to: Abhishek Pratap Singh <abhishek_s@cs.iitr.ac.in>.

*Proceedings of the 43$^{rd}$ International Conference on Machine Learning*, Seoul, South Korea. PMLR 306, 2026. Copyright 2026 by the author(s).

the architectural level (Hao et al., 2023).

- **Semantic Grounding:** A model might correctly predict 'Positiv' affect, but its latent representations often lack alignment with human linguistic concepts. The lack of explicit semantic consistency limits the interpretability of the decision-making process.

To bridge the gap, we introduce the **VIBE** framework. Unlike purely statistical models, VIBE adopts a kinematics-aware architecture that explicitly models the interplay between low-level sensory signals and high-level physical constraints. Our core idea is that for a model to learn effectively, it must do two things: cleanly separate the true emotional signals from background noise, and ensure those signals align with meaningful concepts. We propose three key innovations:

- **Causal Disentanglement:** We employ a Dual-Stream Variational Information Bottleneck (VIB) to separate affective signals from environmental confounders strictly. Unlike standard VIB, we introduce a soft *Orthogonality Constraint* that geometrically forces the "Agent" and "Context" latent spaces to remain disjoint, ensuring the model focuses on agent dynamics rather than background noise.

- **Gated Transformer:** We introduce a Transformer equipped with *Decoupled Adaptive Layer Normalization (AdaLN)*. This module modulates neural attention based on a raw physical synchrony metric ($\gamma$), allowing the model to dynamically shift its processing strategy between "Global Coherence" (high synchrony) and "Local Outliers" (low synchrony) based on the crowd's physical state.

- **Text-Guided Semantic Alignment:** We integrate a Semantic Alignment objective to regularize the latent representation. By projecting the learned features into a shared embedding space, we enforce cosine similarity with pre-trained textual anchors, explicitly grounding the abstract visual dynamics to resolve semantic ambiguity.

By synthesizing the descriptive power of deep neural networks with the structural guidance of kinematics and semantics, VIBE offers a robust, interpretable, and theoretically grounded solution to GER.

## 2. Related Work

This section reviews literature on (a) Deep Learning for GER, (b) Multimodal Fusion and Cross-Attention in GER, and (c) Disentanglement-based approaches in Emotion Recognition (ER), highlighting the research gaps thereof.

### 2.1. Deep Learning for GER

Early approaches to GER relied on hand-crafted descriptors that aggregated facial expressions, audio descriptors, and scene context (Thuseethan et al., 2022; Gong et al., 2025; Sharma et al., 2021). With the advent of multimodal *in-the-wild* datasets like VGAF (Sharma et al., 2019; 2021) and GEVC (Quach et al., 2022), the paradigm shifted toward deep discriminative learning. Hybrid networks, such as those combining CNNs for spatial features and LSTMs for temporal dynamics (Pinto et al., 2020), became the baseline. More recently, Transformer-based architectures have dominated; for instance, (Waligora et al., 2024) proposed a Joint Multimodal Transformer that fuses audio-visual modalities via localized self-attention. Similarly, (Li et al., 2023) used a multi-task learning framework to aggregate local and global features, while (Huang & Xu, 2025) proposed a spatiotemporal transformer to fuse information across three modalities effectively. Complementing these architectural advances, (Liu et al., 2025) introduced a hybrid contrastive learning strategy that leverages bidirectional learning and adaptive mixture models to improve data efficiency and enforce semantic consistency across affective classes.

### 2.2. Multimodal Fusion and Cross-Attention in GER

Effective fusion of audio, visual, and social cues is central to GER (Dhall et al., 2020; Jin et al., 2020; Petrova et al., 2020; Dhall et al., 2023). The current state of the art relies heavily on Cross-Modal Attention. (Wang et al., 2020) introduced an implicit knowledge injection strategy, while (Praveen & Alam, 2024) proposed a Recursive Joint Cross-Modal Attention (RJCMA) to capture inter-modal dependencies. These mechanisms allow visual tokens to attend to acoustic features (and vice versa) to resolve ambiguity. While (Kumar et al., 2024a) proposed a decision-fusion-based approach that combines features from four modalities to make the final decision. (Li et al., 2025) proposed FEHSS, a three-subspace framework that assigns soft emotion memberships to individuals while jointly optimizing conflict suppression, tendency regulation, and resonance enhancement through an adaptive hierarchical reinforcement optimizer.

### 2.3. Disentanglement based approaches in Emotion Recognition (ER)

Disentangled Representation Learning (DRL) has emerged as a powerful paradigm for learning robust and interpretable features by separating the underlying factors of data variation into disjoint semantic subspaces (Wang et al., 2024). In the context of Multimodal Emotion Recognition (MER), DRL addresses the inherent heterogeneity of sensory streams by isolating *modality-invariant* (consistency) features from *modality-specific* (complementarity) information.

Standard approaches typically project audio, visual, and tex-

tual modalities into two distinct latent subspaces: a shared subspace capturing standard emotional semantics across all views, and private subspaces encoding modality-unique characteristics. Yang et al. (Yang et al., 2022) leveraged this dual-stream architecture to enhance cross-modal fusion by explicitly reducing redundancy. Building on this foundation, Sun et al. (Sun et al., 2023) introduced the Fine-grained Disentangled Representation Learning (FDRL) framework. FDRL refines the separation process by employing fine-grained alignment and disparity constraints, ensuring that the shared subspace strictly captures consistency while the private subspaces maximize diversity.

However, recent research suggests that the binary distinction between "shared" and "private" is insufficient for handling real-world noise. Zhou et al. (Zhou et al., 2025) argued that modality-specific representations often contain conflicting or task-irrelevant information that degrades predictive performance. To address this, they proposed *TriDiRA*, a triple disentanglement strategy that decomposes the input into three components: modality-invariant, *effective* modality-specific, and *ineffective* modality-specific representations. By filtering out the ineffective components during fusion, this method achieves superior robustness against environmental interference.

**2.4. Inference Summary**
Despite achieving high accuracy on standard benchmarks, current methodologies typically overlook the physical reality of group dynamics, treating them instead as latent statistical patterns rather than grounded physical processes. By lacking explicit mechanisms to model Group Cohesion or physical synchrony, these approaches rely on implicit correlations learned from large-scale data. This often results in models overfitting to superficial environmental cues such as lighting rather than capturing the actual behavioral dynamics of the agents.

Beyond physical modeling, standard Cross-Attention fusion remains purely associative, linking modalities through statistical co-occurrence rather than causal interaction. Critically, these 'always-on' architectures lack the selective gating required to suppress unreliable inputs when group behavior becomes asynchronous. Consequently, in complex scenes where physical coordination is weak, such methods inadvertently amplify environmental noise instead of isolating the underlying affective signal.

Finally, existing disentanglement frameworks in affective computing primarily address *modal heterogeneity*, focusing on separating modality-invariant features from modality-specific ones. However, they fail to address the challenge of *causal interference*. By not explicitly disentangling the *Agent's Affect* from the *Environmental Context*, these frameworks allow irrelevant background details to pollute the emotional representation. This entanglement prevents the model from learning robust, generalized patterns, leading to significant performance degradation in unconstrained, real-world video environments.

## 3. Methodology

### 3.1. Problem Formulation
We consider a multimodal dataset $\mathcal{D} = \{(\mathcal{V}_i, \mathcal{A}_i, \mathcal{T}_i, \mathbf{y}_i)\}_{i=1}^N$. Each sample comprises a video sequence $\mathcal{V}$, a synchronized audio stream $\mathcal{A}$, and a textual description $\mathcal{T}$. The scene contains $K$ interacting agents, and the objective is to predict the group-level valence $\mathbf{y} \in \Delta^{C-1}$ (Positive, Negative, Neutral).

Standard discriminative models approximate the posterior $P(\mathbf{y}|\mathcal{V}, \mathcal{A})$ directly. However, we posit that the observed signal is a result of interacting but distinct factors: Causal Agent Dynamics ($\mathcal{Z}_{aff}$), Environmental Nuisance ($\mathcal{Z}_{env}$), and Global Scene Context ($\boldsymbol{s}$).

The VIBE framework treats the inference process as a structural causal model. We decompose the input tuple into local agent tubes $\boldsymbol{X}_{loc}$, global scene features $\boldsymbol{X}_{glob}$, acoustic features $\boldsymbol{X}_{aud}$, and a physical synchrony matrix $\gamma$. The following composition of operations defines the inference flow:

$$\boldsymbol{s} = \phi_{scene}(\boldsymbol{X}_{glob}) \quad \text{(Scene Calibration Token)} \quad (1)$$

$$\hat{\mathbf{y}} = f_{\text{CLS}} \circ \Psi_{\text{sat}} \circ \Phi_{\gamma,\boldsymbol{s}}(\text{VIB}(\boldsymbol{X}_{loc}), \boldsymbol{X}_{aud}) \quad (2)$$

where $\boldsymbol{s}$ serves as a global context conditioner, $\Phi_{\gamma,\boldsymbol{s}}$ represents the gated temporal reasoning engine (modulated by synchrony $\gamma$ and scene calibration token $\boldsymbol{s}$), and $\Psi_{\text{sat}}$ enforces semantic alignment. The overall architecture is shown in Figure 1.

### 3.2. Phase I: Spatiotemporal Signal Extraction
The raw video stream $\mathcal{V}$ is a high-bandwidth signal containing significant sensor noise and temporal redundancy. Phase I serves as a structural pre-processor, decomposing the raw video into identity-preserving *tubes* and extracting the core feature primitives required for the *Gated Transformer*.

#### 3.2.1. TEMPORAL DECIMATION & TRAJECTORY STABILIZATION
We reduce the input sampling rate to 6 Hz to eliminate the computational overhead of standard 30 FPS video. This decimation preserves all relevant behavioral dynamics, as human macro-expressions generally evolve over a duration of 0.5 to 4.0 seconds (Guerdelli et al., 2022). Identity persistence is maintained via *ByteTrack* (Zhang et al., 2022). To mitigate detector jitter, which downstream modules might misinterpret as high-arousal tremor, we apply a Gaussian

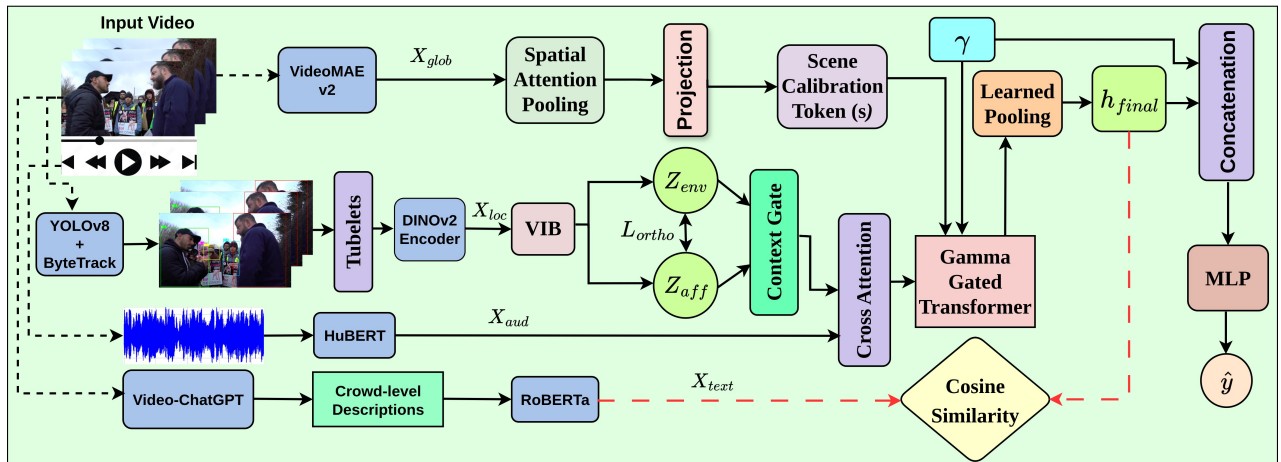

*Figure 1.* Overview of the proposed VIBE architecture. The framework processes features from three pre-trained encoders: Global Context ($X_{glob}$) via VideoMAE V2, Local Agent Tubes ($X_{loc}$) via DINOv2, and Acoustic Prosody ($X_{aud}$) via HuBERT. $\gamma$ denotes the Synchrony Matrix used for dynamic gating, while $X_{text}$ represents the textual anchor features. The Variational Information Bottleneck (VIB) is employed to compress the representation and filter noise, while $\mathcal{L}_{ortho}$ enforces orthogonality between the environmental ($Z_{env}$) and affective ($Z_{aff}$) latent spaces to ensure disentanglement.

smoothing kernel $G$ to the raw trajectory coordinates $\mathcal{T}_{raw}$:

$$\mathcal{T}_{smooth}(t) = (G * \mathcal{T}_{raw})(t) = \sum_{\tau} \mathcal{T}_{raw}(t-\tau) \frac{1}{\sqrt{2\pi}\sigma} e^{-\frac{\tau^2}{2\sigma^2}} \quad (3)$$

acting as a low-pass filter for spatial stability.

### 3.2.2. MULTIMODAL EMBEDDINGS

We extract three distinct feature streams using specialized pre-trained backbones:

- **Global Context ($X_{glob}$):** We utilize VideoMAE V2 (Wang et al., 2023) to capture scene-level dynamics. The backbone outputs a token sequence $X_{glob} \in \mathbb{R}^{T \times 768}$, which the model subsequently pools to generate the Scene Calibration Token ($s$) for affine conditioning.

- **Local Agent Tubes ($X_{loc}$):** For each tracked agent (person), we extract a cropped visual tube and process it via DINOv2 (Oquab et al., 2023). The encoder produces an agent-centric tensor $X_{loc} \in \mathbb{R}^{K \times T \times 768}$, representing the raw actor dynamics.

- **Acoustic Prosody ($X_{aud}$):** To capture paralinguistic cues (pitch, energy) independent of semantics, we employ HuBERT (Hsu et al., 2021), extracting the last hidden state $X_{aud} \in \mathbb{R}^{T \times 768}$.

- **Textual Context ($X_{text}$):** We leverage crowd-level descriptions from the dataset annotations to capture high-level semantic concepts. RoBERTa (Liu et al., 2019) encodes these descriptions into a feature sequence $X_{text} \in \mathbb{R}^{L \times 768}$ (where $L$ denotes the tokenized sequence length), which serves as the linguistic anchor for the semantic alignment objective.

### 3.2.3. THE SYNCHRONY MATRIX

A core innovation of VIBE is the explicit quantification of *Group Cohesion*. We compute the instantaneous velocity vector $\mathbf{v}_k(t)$ for each agent's centroid. We then construct a pairwise Correlation Matrix $\mathbf{S}(t) \in \mathbb{R}^{K \times K}$ representing the directional alignment between agents $i$ and $j$:

$$S_{i,j}(t) = \frac{\mathbf{v}_i(t) \cdot \mathbf{v}_j(t)}{\|\mathbf{v}_i(t)\|\|\mathbf{v}_j(t)\| + \epsilon} \quad (4)$$

Equation 4 matrix is used to derive the scalar *Synchrony Matrix* ($\gamma$), defined as the mean off-diagonal correlation over the active window. This scalar $\gamma \in [-1, 1]$ serves as the gating control signal for the Transformer's AdaLN layers, mathematically representing the group's physical cohesion. A detailed derivation along with a stability analysis is presented in Appendix F.

### 3.3. Phase II: Geometric Feature Isolation via Orthogonal VIB

Standard deep encoders are prone to shortcut learning (Arjovsky et al., 2020), maximizing the mutual information between the latent space and targets by exploiting spurious environmental correlations rather than analyzing agent behavior. To enforce geometric feature isolation, which provides the necessary statistical foundation for our subsequent causal modeling, we implement a Dual-Stream VIB applied specifically to the local agent embeddings $X_{loc}$.

We employ two parallel probabilistic encoders, $E_{aff}$ and $E_{env}$ (refer Fig. 2), to project the local agent features into two distinct latent distributions: the affective cause $\mathcal{Z}_{aff}$ (agent dynamics) and the environmental nuisance $\mathcal{Z}_{env}$ (contextual noise). To enable end-to-end backpropagation through these stochastic distributions, we employ the reparameterization trick. For each stream $k \in \{aff, env\}$, the

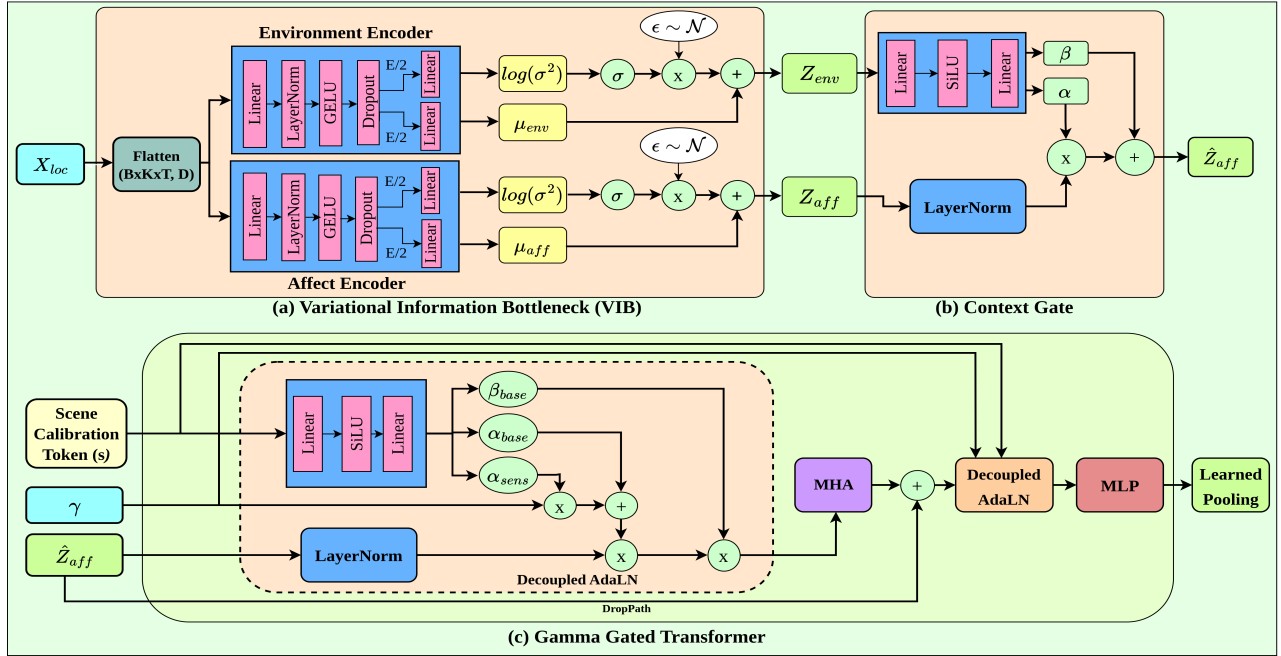

*Figure 2.* Architecture of the (a) Variational Information Bottleneck (VIB), (b) Context Gating Mechanism and (c) Gamma Gated Transformer with Decoupled Adaptive LayerNorm.

latent vector $\mathbf{z}_k$ is sampled as:

$$\mathbf{z}_k = \boldsymbol{\mu}_k(\boldsymbol{X}_{loc}) + \boldsymbol{\epsilon} \odot \boldsymbol{\sigma}_k(\boldsymbol{X}_{loc}), \quad \text{where} \boldsymbol{\epsilon} \sim \mathcal{N}(0, I) \quad (5)$$

We enforce disentanglement via a composite objective that combines information compression via KL divergence and orthogonality via cosine similarity.

### 3.3.1. INFORMATION COMPRESSION ($\mathcal{L}_{KL}$)

We minimize the Kullback-Leibler divergence between the latent posteriors and a spherical Gaussian prior $r(\mathbf{z}) \sim \mathcal{N}(0, I)$. This regularization acts as a low-pass information filter, discarding high-frequency input noise that is not essential for the task:

$$\mathcal{L}_{KL} = \sum_{k \in \{aff, env\}} D_{KL}(q(\mathcal{Z}_k | \boldsymbol{X}_{loc}) || r(z)) \quad (6)$$

### 3.3.2. ORTHOGONALITY CONSTRAINT ($\mathcal{L}_{ortho}$)

To ensure the two latent spaces remain geometrically distinct, we enforce a soft orthogonality constraint. We penalize the squared cosine similarity between the flattened representations of $\mathcal{Z}_{aff}$ and $\mathcal{Z}_{env}$:

$$\mathcal{L}_{ortho} = \frac{1}{B} \sum_{i=1}^{B} \|\text{CosSim}(\text{flat}(\mathcal{Z}_{aff}^{(i)}), \text{flat}(\mathcal{Z}_{env}^{(i)}))\|^2 \quad (7)$$

This forces the "Agent" and "Context" vectors to be orthogonal in the hypersphere, ensuring that $\mathcal{Z}_{aff}$ captures purely behavioral dynamics unpolluted by the static environment.

The VIB and orthogonality constraints function primarily as geometric regularizers for feature isolation. By enforc-

ing this spatial separation, they establish the decoupled statistical foundation necessary for the framework to model directed causal relationships through the structural interventions detailed in Phase III.

### 3.4. Phase III: Interaction Modeling

The disentangled latents are processed through a multi-stage interaction engine designed to fuse modalities while respecting the physical state of the crowd.

### 3.4.1. LOCAL CONTEXT INJECTION & CROSS-MODAL ALIGNMENT

While VIB enforces geometric orthogonality, the environmental context is still necessary for conditioning. We instantiate the causal nature of the framework through two key mechanisms during this interaction modeling phase. First, the causal relationship is explicitly instantiated by the Context Injection Gate through affine modulation. Rather than treating modalities as symmetric during fusion, we use the context latent $\boldsymbol{\mathcal{Z}}_{env}$ to affine-modulate the affective latent $\boldsymbol{\mathcal{Z}}_{aff}$. This mathematically models a directed Structural Causal Model (SCM), where the environment acts as a causal prior that conditions the behavioral state:

$$\boldsymbol{\mathcal{Z}}_{aff} = Norm(\boldsymbol{\mathcal{Z}}_{aff}) \odot (1 + \phi(\boldsymbol{\mathcal{Z}}_{env})) + \psi(\boldsymbol{\mathcal{Z}}_{env}) \quad (8)$$

where $\phi$ and $\psi$ are learned projections.

Second, to prove this directed relationship is functional, we conduct Latent Swapping interventions (Section 4.3.4).

Subsequently, we fuse acoustic information using a standard Cross-Attention block, where the refined visual tokens $\hat{\mathcal{Z}}_{aff}$ act as Queries and the acoustic embeddings $\boldsymbol{X}_{aud}$ act as Keys and Values.

### 3.4.2. THE GAMMA-GATED TRANSFORMER

Standard Transformers are isotropic, processing chaotic crowds and synchronized marches identically. VIBE introduces a Decoupled AdaLN that modulates the temporal feature space based on the scalar synchrony metric $\gamma$ and the global scene calibration token $\boldsymbol{s}$.

Unlike standard AdaLN, which predicts affine parameters solely from a static context, we decouple the modulation into a *static baseline* and a *dynamic sensitivity* component. The scene calibration token $\boldsymbol{s}$ is projected into three vectors: baseline scale $\boldsymbol{\alpha}_{base}$, baseline shift $\boldsymbol{\beta}_{base}$, and synchrony sensitivity $\boldsymbol{\alpha}_{sens}$. The final modulation parameters are computed as:

$$\text{Scale}(\gamma, \boldsymbol{s}) = \underbrace{\boldsymbol{\alpha}_{base}(\boldsymbol{s})}_{\text{Static Context}} + (\gamma \cdot \underbrace{\boldsymbol{\alpha}_{sens}(\boldsymbol{s})}_{\text{Sync Sensitivity}}) \qquad (9)$$

$$\text{AdaLN}(\boldsymbol{x}) = \text{Norm}(\boldsymbol{x}) \odot (1 + \text{Scale}(\gamma, \boldsymbol{s})) + \boldsymbol{\beta}_{base}(\boldsymbol{s}) \qquad (10)$$

This mechanism allows the model to learn specific sensitivity profiles for different group dynamics.

### 3.5. Phase IV: Semantic Alignment & Classification

The final phase transforms the temporal features into a decision vector. Crucially, rather than mapping directly to class labels, which yields a "black box" decision, we enforce an intermediate step of Semantic Alignment.

### 3.5.1. TEMPORAL POOLING

The sequence of $\gamma$ modulated features $\boldsymbol{H}_{seq} \in \mathbb{R}^{T \times D}$ is aggregated into a single video-level representation $\boldsymbol{h}_{final}$ using *Learned Query Attention*. Unlike mean pooling, which treats all frames equally, this mechanism learns a static query vector (q) to compute a weighted sum of the temporal states, allowing the model to prioritize "key frames" where the group dynamic is most salient:

$$\boldsymbol{h}_{final} = \sum_{t=1}^{T} \alpha_t \boldsymbol{H}_{seq}(t) \qquad (11)$$

where $\alpha_t = \text{softmax}(\boldsymbol{H}_{seq}(t)^T \cdot q)$

### 3.5.2. VISION-TEXT ALIGNMENT

To ground the latent representation $\boldsymbol{h}_{final}$ in human-understandable concepts, we project it into a Rationale Space aligned with the linguistic manifold. We employ a linear projector $P_{sat}$ to map $\boldsymbol{h}_{final}$ to $\boldsymbol{z}_{rat}$ (the visual rationale). During training, we utilize the textual descriptions generated via Video-ChatGPT (Maaz et al., 2024). We encode these descriptions using a frozen pre-trained RoBERTa

model to obtain Text Anchors $\boldsymbol{z}_{text}$. We then maximize the cosine similarity between the visual rationale and the text anchor. This forces the visual encoder to organize the latent space according to semantic concepts without requiring explicit manual attribute labeling.

### 3.5.3. CLASSIFICATION

For the final prediction, we explicitly reintroduce the scalar synchrony metric $\bar{\gamma}$ (averaged over time) to the semantic features. This ensures the classifier has direct access to the crowd's raw physical state. The fused vector is passed through an MLP to predict the final valence probability $\hat{\mathbf{y}}$.

### 3.6. Loss Function

The VIBE framework is trained end-to-end using a composite objective function that handles label ambiguity, enforces causal disentanglement, and aligns visual dynamics with semantic concepts.

### 3.6.1. HIERARCHICAL LABEL SMOOTHING ($\mathcal{L}_{HCE}$)

Group emotion labels are inherently noisy and ordinal. A misclassification between 'Positive' and 'Negative' is qualitatively worse than between 'Positive' and 'Neutral'. To embed this ordinal structure, we employ a *Hierarchical Cross-Entropy* loss. We define a semantic adjacency matrix $A \in \mathbb{R}^{3 \times 3}$ (see Appendix C.1.1 for the definition) that distributes probability mass to adjacent classes. Let $\mathbf{y}$ be the one-hot ground truth. We compute smoothed soft targets $\mathbf{y}_{soft} = \mathbf{A}\mathbf{y}$. The loss is computed as the Kullback-Leibler divergence between the predicted logits and these smoothed targets. Additionally, to improve decision boundary robustness, we employ *Manifold Mixup* during training, where the loss is calculated as a convex combination of mixed targets:

$$\mathcal{L}_{HCE} = \lambda \cdot \text{CE}(\hat{\mathbf{y}}, \mathbf{y}_a) + (1 - \lambda) \cdot \text{CE}(\hat{\mathbf{y}}, \mathbf{y}_b) \qquad (12)$$

where $\lambda \sim \beta(\alpha, \alpha)$ is the mixup coefficient.

### 3.6.2. SEMANTIC ALIGNMENT

To ground the latent representation in human-understandable concepts, we use the textual descriptions generated for each dataset using Video-ChatGPT. We project the final temporal embedding $\boldsymbol{h}_{final}$ into a *Rationale Vector* $\boldsymbol{z}_{rat}$ and minimize the *Cosine Distance* between it and the pre-computed RoBERTa embedding of the descriptions $\boldsymbol{X}_{text}$:

$$\mathcal{L}_{sat} = 1 - \frac{\boldsymbol{z}_{rat} \cdot \boldsymbol{X}_{text}}{\|\boldsymbol{z}_{rat}\| \|\boldsymbol{X}_{text}\|} \qquad (13)$$

This effectively pulls the neural features towards the linguistic manifold, ensuring that the features used for classification are semantically consistent with the scene description.

### 3.6.3. STRUCTURAL REGULARIZATION

To enforce the structural constraints of the VIB module, we minimize two auxiliary losses. First, we compute the

*Table 1.* Table presents a comparison of existing approaches on the VGAF and GECV datasets. The results demonstrate that the proposed approach, *VIBE*, outperforms existing methods. The best results are shown in **bold**. S: Scene, A: Audio, F: Face, T: Text.

| Dataset | Reference | Methodology | Modalities | Acc.↑ |
|---|---|---|---|---|
| VGAF | (Sharma et al., 2019) | FC-LSTM | S + A | 50.23% |
| | (Wang et al., 2020) | K-injection network | S + A | 63.25% |
| | (Pinto et al., 2020) | Activity Recognition Network (ARN) | S + A | 61.83% |
| | (Sharma et al., 2021) | VGAFNet | F + S + A | 61.61% |
| | (Evtodienko, 2021) | Cross-modal attention | S + A | 60.37% |
| | (Shvetsova et al., 2022) | Everything at once | S + F | 45.93% |
| | (Dhall et al., 2023) | InceptionV3+LSTM | S + A | 51.30% |
| | (Li et al., 2023) | CLS+MSE | S + F | 68.41% |
| | (Waligora et al., 2024) | MMER | S + F | 52.23% |
| | (Praveen & Alam, 2024) | RJCMA | S + F | 47.51% |
| | (Huang & Xu, 2025) | MSST | S + F | 66.42% |
| | **Proposed** | VIBE | A + S + T | **70.17%** |
| GECV | (Quach et al., 2022) | TNVPF | F | 70.97% |
| | (Shvetsova et al., 2022) | Everything at once | S + F | 81.93% |
| | (Waligora et al., 2024) | MMER | S + F | 80.49% |
| | (Praveen & Alam, 2024) | RJCMA | S + F | 53.66% |
| | (Huang & Xu, 2025) | MSST | S + F | 86.87% |
| | **Proposed** | VIBE | A + S + T | **91.85%** |

KL divergence ($\mathcal{L}_{KL}$) between the latent posteriors and a standard normal prior $\mathcal{N}(0, I)$ to regularize the embedding space. Second, we enforce geometric disentanglement by minimizing the squared cosine similarity ($\mathcal{L}_{ortho}$) between the flattened Affective ($\mathcal{Z}_{aff}$) and Environmental ($\mathcal{Z}_{env}$) latents, as defined in Sec. 3.3.

### 3.6.4. TOTAL OBJECTIVE
The final objective function is a weighted sum of the task loss ($L_{HCE}$) and structural regularizers ($L_{sat}, L_{ortho}, L_{KL}$):

$$\mathcal{L}_{total} = \mathcal{L}_{HCE} + \lambda_{sat}\mathcal{L}_{sat} + \lambda_{ortho}\mathcal{L}_{ortho} + \mathcal{L}_{KL} \quad (14)$$

In our experiments, we set $\lambda_{sat} = 0.5$ and $\lambda_{ortho} = 0.1$, rounded from the optimal values identified using the Optuna library (Akiba et al., 2019) to balance semantic alignment and classification performance. Appendix C presents an in-depth theoretical analysis of the proposed loss components.

## 4. Results and Discussions

### 4.1. Datasets
In this work, we have used two publicly available datasets: VGAF (Sharma et al., 2021) and GECV (Quach et al., 2022). The VGAF dataset comprises 4,183 samples (2,661 training, 766 validation, and 756 testing), with 5-second labeled clips extracted from videos of varying resolutions and frame rates (13–30 fps). On the other hand, the GECV dataset consists of 627 longer sequences (10–20 seconds, approx. 300 frames) where each frame strictly captures interactions between multiple individuals. More details on the data characteristics, preprocessing, and experimental setup are

provided in Appendix B.

*Table 2.* Ablation study on feature aggregation strategies. *GAP*: Global Average Pooling; *Naive Concat*: Simple channel-wise concatenation; *Standard AdaLN*: Adaptive Layer Norm with only scene calibration token as gating.

| Dataset | Concatenation Method | Acc.↑ | F1↑ |
|---|---|---|---|
| VGAF | GAP | 68.98% | 0.691 |
| | Naive Concat | 66.52% | 0.665 |
| | Standard AdaLN | 68.20% | 0.682 |
| | Proposed | **70.17%** | **0.701** |
| GECV | GAP | 90.20% | 0.901 |
| | Naive Concat | 88.56% | 0.884 |
| | Standard AdaLN | 91.82% | **0.916** |
| | Proposed | **91.85%** | 0.915 |

### 4.2. Comparative Analysis
Table 1 presents a comprehensive comparison of VIBE against state-of-the-art methods, establishing a new benchmark for group affect prediction with accuracies of **70.17%** on VGAF and **91.85%** on GECV. On the challenging VGAF dataset, our framework outperforms the strong baseline MSST (Huang & Xu, 2025) by 3.75%, while methods relying solely on Scene and Face modalities, such as Everything-at-Once (Shvetsova et al., 2022), struggle significantly (45.93%), likely due to the unconstrained background noise. By integrating textual anchors ($T$) with strictly enforced physical consistency, VIBE effectively resolves these ambiguities. This capability is even more pronounced on the GECV dataset, where we achieve a substantial +4.98% gain over MSST. This sharp performance jump suggests that our *Synchrony Matrix* mechanism is particularly effective

in longer, denser crowds where collective physical coordination serves as a more robust proxy for group valence than isolated facial expressions.

## 4.3. Ablation Studies

We perform extensive ablation studies to validate the contribution of each individual component.

### 4.3.1. IMPACT OF FUSION STRATEGY

Table 2 evaluates distinct mechanisms for integrating Global Context ($X_{glob}$) with Local Agent dynamics, highlighting the limitations of conventional aggregation methods when scaling to larger, unconstrained datasets. A consistent failure is observed with *Naive Concat*, which yields the lowest accuracy across benchmarks, confirming that merely appending high-dimensional scene features allows environmental noise to drown out subtle affective signals.

Furthermore, intermediate baselines reveal a critical divergence based on data scale and complexity. On the smaller GECV dataset, *Standard AdaLN* achieves strong performance, even yielding a marginally higher F1 score than our proposed method. However, as environmental diversity increases, this static context prior becomes a liability. On the larger, more complex VGAF dataset, *Standard AdaLN* collapses, falling behind even *GAP*. This highlights that in highly diverse scenes, a static, unmodulated environmental prior becomes distracting rather than helpful. The Proposed method resolves this scalability and robustness issue. While maintaining near-identical performance on GECV, it drives a substantial recovery on VGAF (+1.97% accuracy over *Standard AdaLN*). This explicitly validates the role of the Synchrony metric ($\gamma$). By leveraging real-time $\gamma$ to adaptively gate environmental influence, VIBE prevents context distraction and maintains stable, robust performance even in the highly diverse scenes characteristic of large-scale, real-world datasets.

### 4.3.2. IMPACT OF LOSS OBJECTIVES

Table 3 isolates the contribution of each training objective, revealing a critical dependency between geometric regularization and semantic grounding. A key finding is that geometric constraints ($\mathcal{L}_{ortho}, \mathcal{L}_{KL}$) are detrimental when applied in isolation. For instance, the configuration utilizing only disentanglement priors ($\mathcal{L}_{HCE} + \mathcal{L}_{ortho} + \mathcal{L}_{KL}$) degrades performance below the baseline on both datasets (dropping to 66.70% on VGAF and 89.39% on GECV). This confirms that forcing feature separation without semantic guidance results in arbitrary manifold fragmentation, thereby reducing discriminative power.

Furthermore, the results highlight the stabilizing role of the Orthogonality Constraint. On the GECV dataset, removing $\mathcal{L}_{ortho}$ while retaining the Information Bottleneck ($\mathcal{L}_{HCE} + \mathcal{L}_{KL} + \mathcal{L}_{sat}$) causes a sharp performance drop

Table 3. Ablation study on loss functions. *Text Alignment*: Semantic alignment with text anchors; *HCE*: Hierarchical Cross Entropy.

| Dataset | Method | Acc. ↑ | F1 ↑ |
|---|---|---|---|
| VGAF | $L_{HCE}$ | 67.33% | 0.673 |
| | $L_{HCE} + L_{ortho} + L_{KL}$ | 66.70% | 0.666 |
| | $L_{HCE} + L_{ortho} + L_{sat}$ | 67.67% | 0.675 |
| | $L_{HCE} + L_{KL} + L_{sat}$ | 67.41% | 0.673 |
| | $L_{HCE} + L_{ortho}$ | 67.27% | 0.665 |
| | $L_{HCE} + L_{sat}$ | 68.41% | 0.684 |
| | $L_{HCE} + L_{KL}$ | 66.36% | 0.664 |
| | Proposed Loss | **70.17%** | **0.701** |
| GECV | $L_{HCE}$ | 90.97% | 0.907 |
| | $L_{HCE} + L_{ortho} + L_{KL}$ | 89.39% | 0.892 |
| | $L_{HCE} + L_{ortho} + L_{sat}$ | 91.66% | **0.917** |
| | $L_{HCE} + L_{KL} + L_{sat}$ | 87.74% | 0.874 |
| | $L_{HCE} + L_{ortho}$ | 90.64% | 0.904 |
| | $L_{HCE} + L_{sat}$ | 90.18% | 0.900 |
| | $L_{HCE} + L_{KL}$ | 90.01% | 0.887 |
| | Proposed Loss | **91.85%** | 0.915 |

to 87.74%. This suggests that without the structural barrier of orthogonality, the VIB's aggressive compression over-regularizes the representation, discarding useful signals.

In contrast, the full *Proposed Loss* achieves the highest accuracy across both benchmarks (**70.17%** on VGAF and **91.85%** on GECV). The significant jump in performance compared to the semantic-only variant ($\mathcal{L}_{HCE} + \mathcal{L}_{sat}$) demonstrates a powerful synergistic effect: $\mathcal{L}_{sat}$ provides the necessary semantic map, while $\mathcal{L}_{ortho}$ and $\mathcal{L}_{KL}$ refine the structural topology. Ultimately, all components are essential; removing any single loss disrupts the delicate balance between disentanglement, compression, and semantic alignment.

Table 4. Sensitivity analysis on the number of tracked agents ($K$).

| Dataset | No of Agents | Acc. ↑ | F1 ↑ |
|---|---|---|---|
| VGAF | 3 | 67.80% | 0.678 |
| | 5 | 69.38% | 0.694 |
| | 8 | **70.17%** | **0.701** |
| | 10 | 68.88% | 0.667 |
| GECV | 3 | 90.65% | 0.892 |
| | 5 | 91.81% | **0.916** |
| | 8 | **91.85%** | 0.915 |
| | 10 | 91.82% | 0.915 |

### 4.3.3. SENSITIVITY TO CROWD DENSITY

Table 4 investigates the sensitivity of the model to the crowd capacity parameter ($K$), revealing a crucial trade-off between maximizing social context and minimizing peripheral noise. Performance consistently peaks at $K = 8$ across both benchmarks (**70.17%** on VGAF and **91.85%** on GECV), indicating that this threshold effectively captures the primary *affective circle* of the interaction without over-sampling.

Beyond this point, different characteristics of the dataset emerge: in the complex VGAF dataset, increasing $K$ to 10 results in a sharp drop in precision (to 68.88%), confirming that forcing peripheral actors to include introduces irrelevant behavioral noise that dilutes the group signal. In contrast, performance on GECV remains remarkably stable at higher densities ($K = 10$). It achieves near-peak results even with sparse tracking ($K = 5$, 91.81%), demonstrating VIBE's ability to robustly infer group valence even when the entire social graph is partially missing or limited.

### 4.3.4. VALIDATION OF DISENTANGLEMENT

We visualize the penultimate layer embeddings using t-SNE (Figure 4) to understand the topological structure of the feature space. Fusion baselines, such as GAP and Naive Concatenation, exhibit diffuse clustering and significant class overlap. This visually confirms that static aggregation fails to filter environmental noise, leading to ambiguous decision boundaries. In contrast, the full VIBE framework demonstrates a robust topology with sharp intra-class compaction and clear inter-class margins. This visual distinction validates that our disentanglement mechanisms successfully suppress irrelevant scene features, thereby separating the latent spaces.

Beyond qualitative visualizations, we quantitatively confirm the separation of the representations $Z_{aff}$ and $Z_{env}$ (Appendix D.1). The Pearson correlation between $Z_{aff}$ and $Z_{env}$ is effectively zero, which confirms geometric independence. Furthermore, the Hilbert-Schmidt Independence Criterion (HSIC) approaches zero, mathematically guaranteeing non-linear statistical independence between the causal and contextual latent spaces. To ensure that the model does not rely on identity shortcuts or background bias, we performed linear probing on frozen encoders. The classification accuracy of the $Z_{env}$ probe collapses to 36.01%, confirming that affective shortcuts have been successfully removed from the environmental context. Meanwhile, the $Z_{aff}$ probe achieves 55.72% in isolation, demonstrating its sensitivity to genuine expressions prior to downstream cross-modal fusion.

While the environment is decoupled from the causal emotion, it is not simply discarded; it is reinjected as a powerful conditional prior. To prove that this directed structural relationship is functionally stable, we conducted a Latent Swapping test. By isolating $Z_{aff}$ and injecting a fundamentally contrasting $Z_{env}$ into test samples, we observed a bounded, average shift in the output probability of $\Delta y = 0.2412$. If features were functionally entangled, swapping these latents would cause a catastrophic distribution shift and complete misclassification. Instead, this bounded 24.12% variance proves that the local context acts as a modular conditional prior, while $Z_{aff}$ correctly retains the primary predictive authority, ensuring the framework remains expression-centric.

## 5. Limitations and Risks for Privacy

### 5.1. Limitations

Although VIBE achieves strong predictive performance, it presents a few limitations. The model is sensitive to the crowd capacity parameter ($K$), as over-sampling in dense scenes can introduce peripheral noise. Additionally, performance is tied to tracking reliability, rapid motion or occlusions leading to identity switches can disrupt both trajectory and synchrony estimations. Lastly, the 6 Hz temporal decimation may preclude the detection of micro-expressions.

### 5.2. Risks Related to Privacy

Group-level emotion recognition systems can raise concerns related to privacy and potential biases. In particular, inferring emotional states from visual and behavioral cues may introduce privacy considerations if applied without appropriate consent or safeguards. Additionally, as with other data-driven approaches, there is a possibility of bias due to imbalanced or non-representative training data, such as the limited scale of existing datasets, which may affect fairness across different social contexts. To mitigate some of these concerns, VIBE is designed to learn behavior-centric representations, rather than relying on identity-specific or sensitive attributes. This design choice reduces reliance on potentially sensitive cues, however, it does not fully eliminate biases that may arise from the data itself. We emphasize that VIBE is intended as a research-oriented framework to advance group-level behavioral understanding, rather than for direct deployment in high-stakes or sensitive real-world applications. Any practical use would require careful consideration of fairness, privacy, and contextual factors, along with appropriate safeguards such as ethical guidelines, data protection measures, and human oversight.

## 6. Conclusion

This paper presents VIBE, a new approach to Group Emotion Recognition that prioritizes how people move and interact with the static details of their environment. While previous models often struggle with "background noise" mistakenly using the setting of a scene to guess the mood, VIBE uses a kinematics-aware framework to isolate genuine human behavior. By introducing Decoupled AdaLN, we successfully separated emotional cues from environmental clutter, allowing the model to focus on physical coordination and group synchrony. The results of our experiment show that when a model understands the physical "Vibe" of a crowd, it performs significantly better than models that simply look at faces in isolation. Our results validate that the environmental context should act as a guide, not a distraction. Ultimately, VIBE moves the field of affective computing toward a more grounded, physics-based understanding of social dynamics.

## Impact Statement

This work shows that causal constraints combined with behavior-centric representations substantially improve AI's ability to model complex social environments. By separating human behavior from the surrounding environment, our framework (VIBE) prevents "shortcut learning" a common problem where models make guesses based on background details or location instead of focusing on how people are actually acting. This approach has two major benefits for society:

- **Fairness and Robustness:** Because VIBE ignores environmental distractions, it helps prevent biased or misleading predictions that can happen when an AI misinterprets a specific setting. This makes crowd analysis and surveillance systems more equitable and reliable across diverse, real-world locations.

- **Responsible AI Development:** By focusing on the "why" behind human movement (causal reasoning), our work provides a foundation for more transparent and ethical AI. It ensures that multimodal models are less prone to errors caused by context and are better aligned with actual human behavior.

While these technologies must always be used within legal and ethical guidelines, this research offers a technical way to reduce unintended harms and make emotion-aware systems more accountable.

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

# A. Representative Examples Showing Ground Truth and Model Predictions for Group Emotion Recognition.

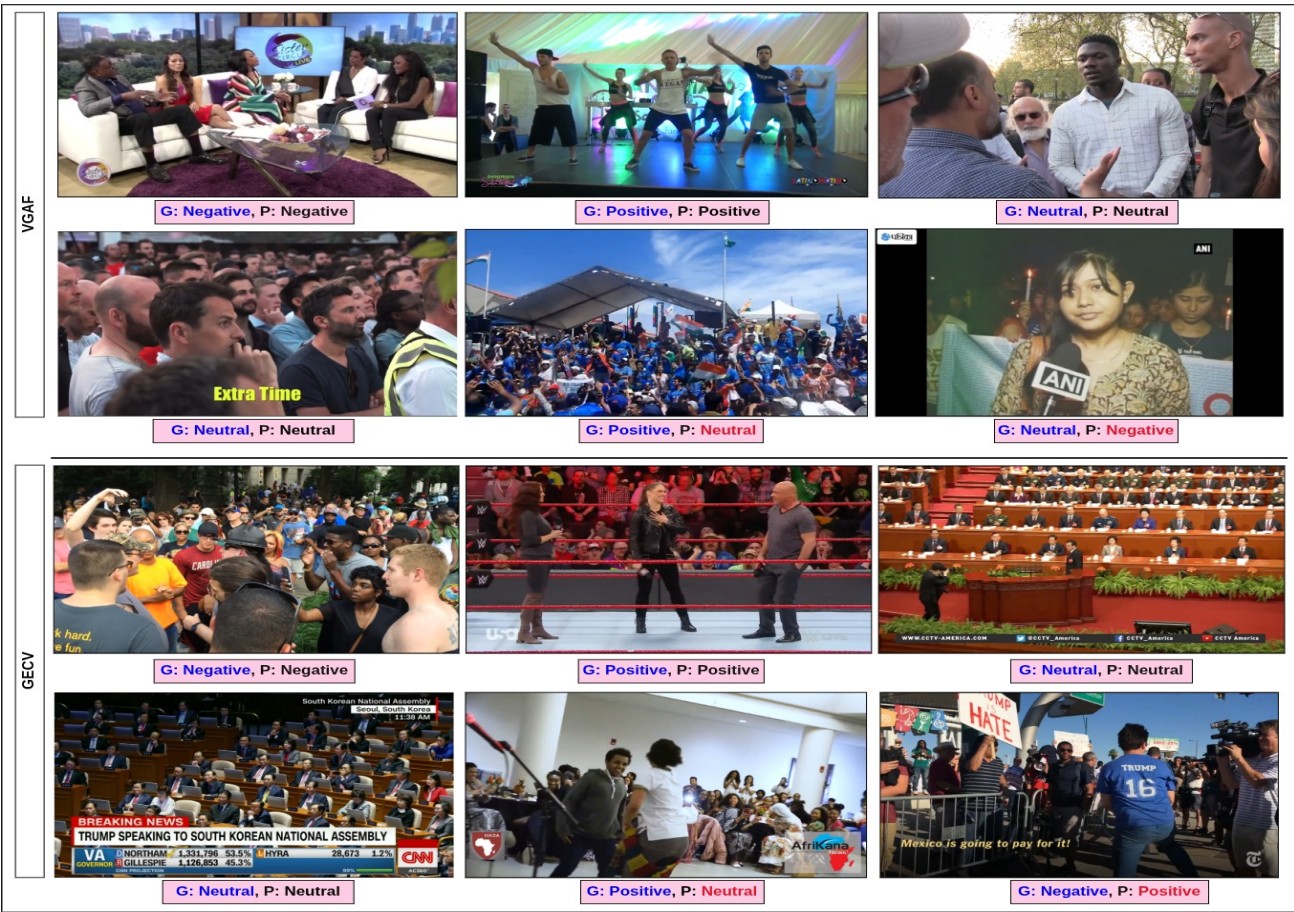

*Figure 3.* Diagram illustrating sample video frames from the VGAF and GECV datasets, along with ground-truth labels and VIBE model predictions. Here, G denotes Ground Truth and P denotes Predictions.

# B. Experimental Setup

To ensure reproducibility, we detail the specific hardware infrastructure, software environment, and hyperparameter configurations used to train the VIBE framework.

### B.1. Implementation Environment

All experiments were conducted on a workstation equipped with an NVIDIA A2000 GPU with 12GB VRAM. The system is powered by an Intel® Xeon® W-2265 CPU and 64 GB of RAM.

### B.2. Training Configuration

The model is trained using the *AdamW* optimizer with a weight decay of $1 \times 10^{-3}$ to prevent overfitting. We employ a Cosine Annealing learning rate scheduler with a warm-up period of 3 epochs. Additional details regarding the training pipeline are presented in Table 5.

### B.3. Computational Characteristics and Efficiency

To ensure complete transparency regarding deployment feasibility, we analyzed the computational profile of both the isolated VIBE model and the full end-to-end inference pipeline which includes the extraction overhead from pre-trained backbones

*Table 5.* Detailed specification of the hyperparameter configuration employed for the VIBE framework. The table outlines the optimization settings, transformer architecture specifics, and the weighting coefficients ($\lambda$) used to balance the auxiliary loss terms.

| Hyperparameter | Value |
|---|---|
| Batch Size | 16 |
| Total Epochs | 30 |
| Initial Learning Rate | $1 \times 10^{-4}$ |
| Latent Dimension ($D$) | 512 |
| Transformer Heads | 8 |
| Transformer Depth | 2 Layers |
| Dropout Rate | 0.3 |
| Mixup Alpha ($\alpha$) | 0.2 |
| Loss Weight $\lambda_{sat}$ | 0.5 |
| Loss Weight $\lambda_{ortho}$ | 0.1 |

such as VideoMAE V2, DINOv2, and HuBERT. The evaluation was conducted on the VGAF dataset, the full characteristics are presented in Table 6.

The standalone VIBE architecture is highly lightweight, comprising only 16.48M parameters with a storage size of 62.86 MB in FP32. By employing uniform temporal decimation, we prevent the Gamma Gated Transformer from processing redundant frames, which strictly minimizes the computational cost to 1.30 G MACs (2.60 G FLOPs). Consequently, the isolated model demonstrates exceptional speed, achieving a latency of 3.96 ms per clip and a throughput of 252.35 video/s.

When incorporating the heavy spatiotemporal feature extraction required by the raw input modalities, the total computational cost of the full pipeline rises to 857.89 G MACs. This yields a full-system latency of 816.1 ms per clip (a throughput of 1.23 video/s). Despite the reliance on multiple powerful feature extractors, the overall memory footprint remains highly accessible. The complete pipeline requires a peak VRAM of just 1172.2 MB and total system RAM of 1569.6 MB. By actively downsampling the input sequence to 6 Hz, VIBE avoids the exponential memory growth typically associated with 30 FPS video processing, ensuring stable operation without out-of-memory bottlenecks, even on standard mid-range hardware.

*Table 6.* Computational metrics for the standalone VIBE model and the full inference pipeline on the VGAF dataset.

| Metric | VIBE Model | Full Pipeline |
|---|---|---|
| Parameters | 16.48M | - |
| Size (FP32) | 62.86 MB | - |
| Latency / Clip | 3.96 ms | 816.1 ms |
| Throughput | 252.35 video/s | 1.23 video/s |
| MACs | 1.30 G | 857.89 G |
| FLOPs | 2.60 G | 1715.77 G |
| Peak VRAM | - | 1172.2 MB |
| Total RAM | - | 1569.6 MB |

## B.4. Dataset Statistics and Characteristics

To rigorously evaluate the VIBE framework, we utilize two distinct "in-the-wild" datasets that vary significantly in scale, sequence duration, and crowd density. Both datasets map group-level affect into three ordinal valence categories: Positive, Negative, and Neutral.

- **VGAF:** The Video Group Affect (VGAF) dataset is a large-scale benchmark comprising 4,183 video samples. The dataset is explicitly partitioned into 2,661 training, 766 validation, and 756 testing samples to prevent data leakage. Each sample is a 5-second labeled clip extracted from unconstrained user-generated videos, featuring diverse resolutions and fluctuating frame rates ranging from 13 to 30 fps. Because of its highly unconstrained nature, VGAF serves as the primary benchmark for testing the model's robustness against severe environmental noise, camera motion, and rapid scene transitions.

- **GECV:** The Group Emotion Recognition in Crowd Videos (GECV) dataset focuses on sustained, dense crowd

interactions rather than short, rapid clips. It consists of 627 longer sequences, with durations ranging from 10 to 20 seconds (approximately 300 frames per video). Every frame in GECV strictly captures active interactions between multiple individuals, providing a denser sociological graph for the model to process. This dataset specifically evaluates the framework's ability to maintain physical synchrony tracking over extended temporal windows.

### B.5. Data Processing and Evaluation Protocol

For temporal consistency, we uniformly sample $T = 32$ frames per video clip. To determine the optimal group capacity, we evaluated agent processing limits of $K \in \{3, 5, 8, 10\}$. We selected a final $K_{max} = 8$ as this configuration yielded the best empirical results across both datasets. If a video contains fewer agents than $K_{max}$, we apply zero-padding and utilize the masking strategy described in Section F to ensure the padding does not influence the synchrony computation.

To ensure rigorous benchmarking and fair comparison, we adopt dataset-specific splitting protocols:

- **VGAF:** We employ a split based on unique video IDs on the training set to prevent data leakage, ensuring that clips from the same source video do not overlap between splits. We reserve $10\%$ of this data for internal validation. The final results are reported on the Validation set provided with the dataset, which we utilize as our held-out test set.

- **GECV:** To maintain comparability with prior literature, we perform a randomized $90 : 10$ train-test split. We repeated this procedure for three distinct random seeds and report the average performance across these three splits on the $10\%$ validation partition.

## C. Objective Function and Optimization

The training of our framework is governed by a composite objective function designed to address the specific challenges of group emotion recognition: label ambiguity, causal disentanglement, and semantic grounding. The total loss $\mathcal{L}_{total}$ is a weighted sum of four distinct components:

$$\mathcal{L}_{total} = \mathcal{L}_{HCE} + \mathcal{L}_{KL} + \lambda_{ortho}\mathcal{L}_{ortho} + \lambda_{sat}\mathcal{L}_{sat} \tag{15}$$

where $\lambda_{ortho}$ and $\lambda_{sat}$ are hyperparameters controlling the strength of the structural and semantic regularization.

### C.1. Hierarchical Cross-Entropy with Manifold Mixup ($\mathcal{L}_{HCE}$)

Standard Cross-Entropy (CE) loss operates under the assumption that all classes are mutually exclusive and equidistant in the semantic space. For a ground truth $y$, standard CE lacks ordinal awareness; it treats the polar opposite class (Negative) and the intermediate class (Neutral) as equally incorrect predictions for a Positive sample. However, group emotion recognition has an inherent ordinal structure: Neutral lies between the two affective extremes. To embed this structural knowledge into the optimization landscape, we employ a hybrid strategy combining Hierarchical Label Smoothing and Manifold Mixup.

#### C.1.1. ORDINAL-AWARE LABEL SMOOTHING

We address the "equidistant error" problem by redefining the target distribution. Instead of optimizing towards a Dirac delta distribution (one-hot vector), we optimize towards a smoothed distribution that respects semantic adjacency.

We define a semantic transition matrix $\mathbf{A} \in \mathbb{R}^{3 \times 3}$, where $A_{ij}$ represents the conditional probability $P(\text{label}_i|\text{truth}_j)$ based on semantic proximity:

$$\mathbf{A} = \begin{bmatrix} 0.85 & 0.15 & 0.00 \\ 0.10 & 0.80 & 0.10 \\ 0.00 & 0.15 & 0.85 \end{bmatrix} \tag{16}$$

The zero entries ($A_{13}$ and $A_{31}$) explicitly enforce a "forbidden transition" between Positive and Negative. This acts as a soft constraint: the model is not penalized for maintaining non-zero probability mass on the *Neutral* class when the ground truth is *Positive*, reflecting the ambiguity of subtle expressions. However, predicting the opposite extreme is heavily penalized.

Given a one-hot ground truth $\mathbf{y}$, the new soft target is $\mathbf{y}_{soft} = \mathbf{A}\mathbf{y}$. The base loss becomes the KL Divergence between the predicted log-probabilities $\log \hat{\mathbf{y}}$ and the soft targets:

$$\mathcal{L}_{smooth} = D_{KL}(\mathbf{y}_{soft} || \hat{\mathbf{y}}) = \sum_{c=1}^{C} \mathbf{y}_{soft}^{(c)} \log \left( \frac{\mathbf{y}_{soft}^{(c)}}{\hat{\mathbf{y}}^{(c)}} \right) \tag{17}$$

This keeps the model from getting too sure of itself. It prevents the model from memorizing mistakes in the data labels, which is a common problem in emotion datasets.

### C.1.2. MANIFOLD MIXUP REGULARIZATION

While label smoothing handles output noise, it does not address the sparsity of the input feature space. High-dimensional video representations often lie on a low-dimensional manifold with large "voids" between class clusters. Standard training leaves the decision boundaries in these voids undefined, leading to brittleness.

To mitigate this, we employ *Manifold Mixup*. Unlike standard input-level Mixup, which creates "ghost" images that may look unnatural, Manifold Mixup interpolates deeply within the latent space. For a pair of samples $(x_i, y_i)$ and $(x_j, y_j)$, we compute their latent representations $\mathbf{z}_i, \mathbf{z}_j$ at the output of the encoder. We then construct a virtual sample $\tilde{\mathbf{z}}$:

$$\lambda \sim \beta(\alpha, \alpha) \tag{18}$$

$$\tilde{\mathbf{z}} = \lambda \mathbf{z}_i + (1 - \lambda) \mathbf{z}_j \tag{19}$$

Here, $\alpha \in (0, \infty)$ denotes the concentration hyperparameter of the $\beta$ distribution, controlling the shape of the mixing probability density. In our framework, we set $\alpha = 0.2$, which induces a U-shaped distribution in which $\lambda$ is predominantly sampled near $0$ or $1$. This ensures that the generated virtual sample $\tilde{\mathbf{z}}$ remains semantically grounded in one of the source classes, avoiding the formation of ambiguous, high-entropy representations in the center of the latent manifold. To enforce decision boundary linearity between these grounded states, we then constrain the model to map this virtual sample to the interpolated targets by minimizing the weighted sum of the respective losses:

$$\mathcal{L}_{HCE} = \lambda \cdot \mathcal{L}_{smooth}(f(\tilde{\mathbf{z}}), \mathbf{y}_{soft}^{(i)}) + (1 - \lambda) \cdot \mathcal{L}_{smooth}(f(\tilde{\mathbf{z}}), \mathbf{y}_{soft}^{(j)}) \tag{20}$$

**Regularization Mechanics:**

1. **Linearizing the Manifold:** By forcing the prediction at $\tilde{\mathbf{z}}$ to be the weighted average of the parents, we strictly penalize sharp, non-linear decision boundaries. This flattens the loss landscape in the "voids" between classes.

2. **Synthesizing Hard Negatives:** When mixing a *Positive* sample with a *Negative* sample, the resulting virtual point lies in the "ambiguous" center of the latent space. Forcing the model to handle this ambiguity robustly improves its ability to classify subtle, real-world examples that sit on the decision boundary.

### C.2. Variational Information Bottleneck ($\mathcal{L}_{KL}$)

Standard deterministic encoders often degenerate into memorization machines, retaining pixel-level nuisance variables (e.g., lighting jitter, video compression artifacts) that correlate with the training labels but fail to generalize. To mitigate this, we adopt the *Information Bottleneck* principle. The objective is to learn a latent encoding $\mathbf{z}$ that maximizes the mutual information with the target $\mathbf{y}$ ($I(\mathbf{z}; \mathbf{y})$), while simultaneously minimizing the mutual information with the input $\mathbf{x}$ ($I(\mathbf{z}; \mathbf{x})$).

We approximate this objective using a variational bound. We treat the encoder as a stochastic channel mapping an input $\mathbf{x}$ to a distribution $q(z|x) = N(\mu, \text{diag}(\sigma^2))$. We constrain this posterior to match a fixed standard normal prior $p(z) = N(0, I)$ by minimizing the Kullback-Leibler (KL) Divergence.

The analytic KL loss for a single sample, assuming a diagonal covariance matrix, is derived as:

$$\mathcal{L}_{KL} = D_{KL}\Big(q(\mathbf{z}|\mathbf{x}) \,\|\, p(\mathbf{z})\Big) = -\frac{1}{2} \sum_{j=1}^{D} \underbrace{\left(1 + \log(\sigma_j^2) - \mu_j^2 - \sigma_j^2\right)}_{\text{Component-wise Contribution}} \tag{21}$$

### C.2.1. THE COMPRESSION-PREDICTION TUG-OF-WAR

This loss function creates a sophisticated adversarial dynamic between the task loss ($\mathcal{L}_{HCE}$) and the regularization term ($\mathcal{L}_{KL}$), effectively acting as a learnable low-pass information filter.

**1. The Centering Force ($\mu_j^2$):** The term $-\mu_j^2$ (inside the negative sum) acts as an $L_2$ regularizer on the latent means. It forces the latent manifold to remain compact around the origin. This prevents the "latent explosion" problem, where the model pushes distinct classes arbitrarily far apart to minimize cross-entropy, leading to a sparse, fragmented latent space that generalizes poorly.

**2. The Uncertainty Pressure ($\sigma_j^2 - \log \sigma_j^2$):** This is the critical component for robustness. The function $f(\sigma^2) = \sigma^2 - \log(\sigma^2)$ is minimized when $\sigma^2 = 1$.

- **Penalty on Certainty:** If the encoder tries to be "too precise" (i.e., collapsing the distribution to a point estimate by driving $\sigma^2 \to 0$), the term $-\log(\sigma^2)$ tends toward infinity, incurring a massive penalty.

- **Noise Injection:** To minimize this penalty, the encoder is forced to increase the variance $\sigma^2$ towards 1. This is equivalent to injecting additive Gaussian noise into the representation.

This creates a *Signal-to-Noise Ratio (SNR)* filter. The classification loss wants to encode every detail to reduce error (driving $\sigma \to 0$), while the KL loss fights to destroy details by adding noise (driving $\sigma \to 1$). Only features with strong causal connections to the emotion generate enough "gradient signal" to overcome the KL penalty and justify a lower variance. Conversely, high-frequency nuisance variables lack this predictive power; the KL loss dominates them, forcing their variance to 1 and their mean to 0, effectively pruning them from the representation.

### C.3. Orthogonality Constraint ($\mathcal{L}_{ortho}$)

A critical limitation of standard multimodal encoders is their susceptibility to *shortcut learning*. Without explicit constraints, deep networks act as "lazy" learners, latching onto the easiest available signal to minimize loss. In group emotion recognition, background context is often statically correlated with emotion, serving as a confounder. Consequently, the encoder may ignore the complex, fine-grained agent dynamics in favor of these static environmental cues.

To dismantle these spurious correlations, we introduce an explicit geometric barrier between the *Agent Dynamics* ($\mathbf{z}_{aff}$) and *Environmental Context* ($\mathbf{z}_{env}$) subspaces. We introduced soft orthogonality by minimizing the squared cosine similarity between their flattened latent representations:

$$\mathcal{L}_{ortho} = \frac{1}{B} \sum_{i=1}^{B} \left\| \text{CoSim}(\mathbf{z}_{aff}^{(i)}, \mathbf{z}_{env}^{(i)}) \right\|^2 = \frac{1}{B} \sum_{i=1}^{B} \left( \frac{\mathbf{z}_{aff}^{(i)} \cdot \mathbf{z}_{env}^{(i)}}{\|\mathbf{z}_{aff}^{(i)}\|_2 \|\mathbf{z}_{env}^{(i)}\|_2} \right)^2 \tag{22}$$

#### C.3.1. GEOMETRIC AND STATISTICAL INTERPRETATION

This constraint fundamentally alters the topology of the learned feature space.

**1. Geometric Independence:** The cosine similarity measures the cosine of the angle $\theta$ between two high-dimensional vectors. Minimizing the squared cosine drives $\cos^2(\theta) \to 0$, which implies $\theta \to 90°$. In a high-dimensional vector space, two orthogonal vectors share no projection onto one another. This means that a variation in the magnitude or direction of $\mathbf{z}_{env}$ results in zero scalar change in the projection along $\mathbf{z}_{aff}$. This effectively isolates the affective features from environmental shifts.

**2. Proxy for Statistical Independence:** While true statistical independence requires minimizing mutual information (which is computationally intractable in high dimensions), geometric orthogonality provides a strong linear proxy. If we assume the latent features are zero-centered (enforced by our standard normal priors in VIB), the inner product corresponds to the unnormalized covariance:

$$\mathbb{E}[\mathbf{z}_{aff}^T \mathbf{z}_{env}] \approx \text{Cov}(\mathbf{z}_{aff}, \mathbf{z}_{env}) \tag{23}$$

By forcing this inner product to zero, we are explicitly decorrelating the features. We prevent the environment encoder from leaking information into the Affective embedding, ensuring that $\mathbf{z}_{aff}$ describes only the agents' behavioral dynamics, invariant to the scene they inhabit.

### C.4. Semantic Alignment ($\mathcal{L}_{sat}$)

Deep visual encoders often operate as black boxes, learning discriminative features that maximize classification accuracy but lack interpretability. A model might correctly classify a scene as *Positive* simply by detecting high brightness or rapid

motion, rather than understanding the underlying social interaction. To rectify this, we introduce a *Semantic Alignment Task (SAT)* that grounds the abstract visual vectors in human-understandable concepts.

We leverage the textual descriptions extracted from the Video-ChatGPT model for the respective datasets as proxies for semantic anchors. These descriptions (e.g., *"A group of friends laughing while eating dinner"*) contain the causal reasoning behind the emotion label. We employ a pre-trained frozen RoBERTa model to extract dense linguistic embeddings $X_{text}$ from these descriptions.

To bridge the modality gap, we project the final visual temporal embedding $h_{final}$ into a shared *Rationale Space* via a learnable linear projection $W_{rat}$:

$$\mathbf{z}_{rat} = W_{rat}\mathbf{h}_{final} + b_{rat} \tag{24}$$

We then align this projected visual rationale with the linguistic embedding by minimizing the *Cosine Distance*:

$$\mathcal{L}_{sat} = 1 - \text{CosSim}(\mathbf{z}_{rat}, \mathbf{X}_{text}) = 1 - \frac{\mathbf{z}_{rat} \cdot \mathbf{X}_{text}}{\|\mathbf{z}_{rat}\|_2 \|\mathbf{X}_{text}\|_2} \tag{25}$$

### C.4.1. MANIFOLD ALIGNMENT

This objective function serves two critical mathematical roles:

**1. Injecting Semantic Commonsense (Knowledge Transfer):** Video datasets are often small, whereas Language Models like RoBERTa are trained on a large amount of text tokens and encapsulate vast amounts of "world knowledge" (e.g., knowing that *cake + candles* usually implies *celebration*). By forcing $z_{rat} \approx X_{text}$, we effectively transfer this structured topology from the language manifold to the visual manifold. The visual encoder is constrained to learn features that are not just discriminative, but descriptive.

**2. Directional Regularization via Cosine Similarity:** In high-dimensional embedding spaces, the magnitude of a vector often correlates with model confidence or feature frequency, while the direction encodes the semantic meaning. Euclidean distance is inherently sensitive to magnitude. Consequently, it allows vectors with large norms to disproportionately influence the metric, effectively conflating signal intensity with semantic dissimilarity and obscuring the accurate directional alignment between features.

By adopting *Cosine Distance*, we address this magnitude bias by effectively projecting all embeddings onto a unit hypersphere. This normalization neutralizes the variance in vector norms, isolating the *angular alignment* as the sole measure of similarity. Geometrically, this objective drives the optimization to align the directional orientation of visual features with their corresponding textual anchors, for instance, pulling the visual manifold of "Positive" affect to overlap with the linguistic cluster of concepts like "joy" or "cheer", thereby ensuring that the learned representations are defined by their semantic content rather than their signal intensity.

### C.5. Fine-Grained Category Analysis

To provide a deeper understanding of the benefits and limitations of the proposed disentanglement framework, we conduct a fine-grained, category-specific evaluation. Specifically, we report the class-wise performance and complete confusion matrices of VIBE on both the VGAF and GECV datasets, comparing it against the Global Average Pooling (GAP) baseline. Note that both configurations utilize disentanglement; their precise structural differences are detailed in Section E. For the GECV dataset, we report the metrics from the specific fold that most closely approximates the overall 3-fold cross-validation average.

Tables 7 and 8 detail the Precision, Recall, F1-Score, and prediction distributions across the affective classes for both datasets. The proposed model demonstrates strong performance across both unconstrained (VGAF) and controlled (GECV) datasets, exceeding the baseline methods. On VGAF, VIBE improves the Negative class F1-score from 0.68 (GAP) to 0.75. On GECV, VIBE boosts the Neutral class F1-score significantly from 0.86 (GAP) to 0.96. Disentanglement successfully filters out environmental noise in these scenarios because the active behavioral and kinematic signals (captured via our synchrony metric, $\gamma$) are robust enough to dominate the static background context.

On the VGAF dataset, the Neutral class exhibits the lowest relative performance for both VIBE (F1: 0.66) and GAP (F1: 0.67), indicating it is a fundamentally challenging category. VIBE frequently confuses Neutral with Positive (51 false

*Table 7.* Class-wise Performance and Confusion Matrices on the VGAF Dataset.

| Method | Class (GT) | Precision | Recall | F1-Score | Support | Positive (P) | Neutral (P) | Negative (P) |
|---|---|---|---|---|---|---|---|---|
| VIBE | Positive | 0.76 | 0.66 | 0.70 | 299 | 196 | 78 | 25 |
| | Neutral | 0.65 | 0.68 | 0.66 | 279 | 51 | 189 | 39 |
| | Negative | 0.70 | 0.81 | 0.75 | 183 | 11 | 23 | 149 |
| GAP | Positive | 0.77 | 0.67 | 0.72 | 299 | 199 | 62 | 38 |
| | Neutral | 0.66 | 0.69 | 0.67 | 279 | 46 | 193 | 40 |
| | Negative | 0.63 | 0.73 | 0.68 | 183 | 12 | 38 | 133 |

*Table 8.* Class-wise Performance and Confusion Matrices on the GECV Dataset.

| Method | Class (GT) | Precision | Recall | F1-Score | Support | Neutral (P) | Negative (P) | Positive (P) |
|---|---|---|---|---|---|---|---|---|
| VIBE | Neutral | 0.92 | 1.00 | 0.96 | 11 | 11 | 0 | 0 |
| | Negative | 1.00 | 0.80 | 0.89 | 10 | 0 | 8 | 2 |
| | Positive | 0.90 | 0.95 | 0.93 | 20 | 1 | 0 | 19 |
| GAP | Neutral | 0.90 | 0.82 | 0.86 | 11 | 9 | 1 | 1 |
| | Negative | 0.82 | 0.90 | 0.86 | 10 | 0 | 9 | 1 |
| | Positive | 0.90 | 0.90 | 0.90 | 20 | 1 | 1 | 18 |

negatives, 78 false positives). On GECV, the Negative class yields the lowest relative F1 (0.89 for VIBE, 0.86 for GAP). Ambiguous or low-expressive social situations for example, people sitting quietly during a panel discussion or standing casually in a street interview do not show strong motion or pronounced paralinguistic cues. In such cases, because the model strictly suppresses the environmental context ($\mathcal{Z}_{env}$), it may unintentionally discard useful contextual information that could aid in disambiguating these subtle situations.

## D. Analysis of Feature Manifolds

To provide deeper insight into the internal representations learned by VIBE, we visualize the penultimate layer embeddings using t-Distributed Stochastic Neighbor Embedding (t-SNE) in Figure 4. This visualization corroborates our quantitative findings by revealing the distinct topological structure induced by different architectural choices.

The fusion baselines (GAP, Naive Concat) exhibit significant class overlap and diffuse clustering. This visually confirms our hypothesis that static aggregation fails to filter environmental noise, resulting in a "muddy" latent space where decision boundaries are ambiguous. In contrast, the proposed Decoupled AdaLN mechanism yields sharper separation, validating its ability to dynamically suppress irrelevant scene features.

The ablation plots further illustrate the necessity of the composite objective. Models trained without semantic alignment ($\mathcal{L}_{sat}$) or orthogonality ($\mathcal{L}_{ortho}$) display fragmented clusters with reduced margins. The full VIBE framework demonstrates the most robust topology: tight intra-class compaction and clear inter-class separability. This proves that the synergistic combination of semantic grounding and geometric disentanglement forces the model to learn a highly discriminative manifold, directly explaining the superior performance reported in Table 3.

### D.1. Quantitative Analysis of Disentanglement

To verify that VIBE effectively decouples the Affective ($\mathbf{Z}_{aff}$) and Environmental ($\mathbf{Z}_{env}$) subspaces, we conducted a statistical correlation analysis on the test sets of both benchmarks. If the representations are truly disentangled, they should exhibit negligible linear dependence. We computed the Pearson Correlation Coefficient ($\rho$) between the flattened latent vectors of the two encoders.

As reported in Table 9, the overall correlation between $\mathbf{Z}_{aff}$ and $\mathbf{Z}_{env}$ remains effectively zero across both datasets (**-0.0174** for VGAF and **0.0054** for GECV). Furthermore, the mean absolute per-dimension correlation is restricted to low values (**0.1045** and **0.1651**, respectively). These consistently low metrics confirm that the proposed Orthogonality Constraint ($\mathcal{L}_{ortho}$) successfully forces the encoders to learn statistically independent features regardless of the dataset characteristics.

To substantiate these findings beyond linear correlation, we further evaluated the VGAF dataset using advanced non-linear and causal metrics (detailed in Table 10). First, the Hilbert-Schmidt Independence Criterion (HSIC) yields a near-zero score of **0.0015**, confirming strict non-linear independence between the two subspaces. Second, linear probing demonstrates that the affective subspace ($\mathbf{Z}_{aff}$) preserves strong task-relevant information (achieving **55.72%** accuracy), whereas the environmental subspace ($\mathbf{Z}_{env}$) drops to near-chance levels (**36.01%**). This confirms that affective signals are successfully purged from the nuisance representation. Finally, a causal intervention test reveals a minimal prediction deviation ($\Delta y$) of **0.2412** when swapping the environmental latent $\mathbf{Z}_{env}$ between samples. Collectively, these metrics ensure that the affective representation remains invariant to background shifts, preventing the "leakage" of environmental context into the agent dynamics.

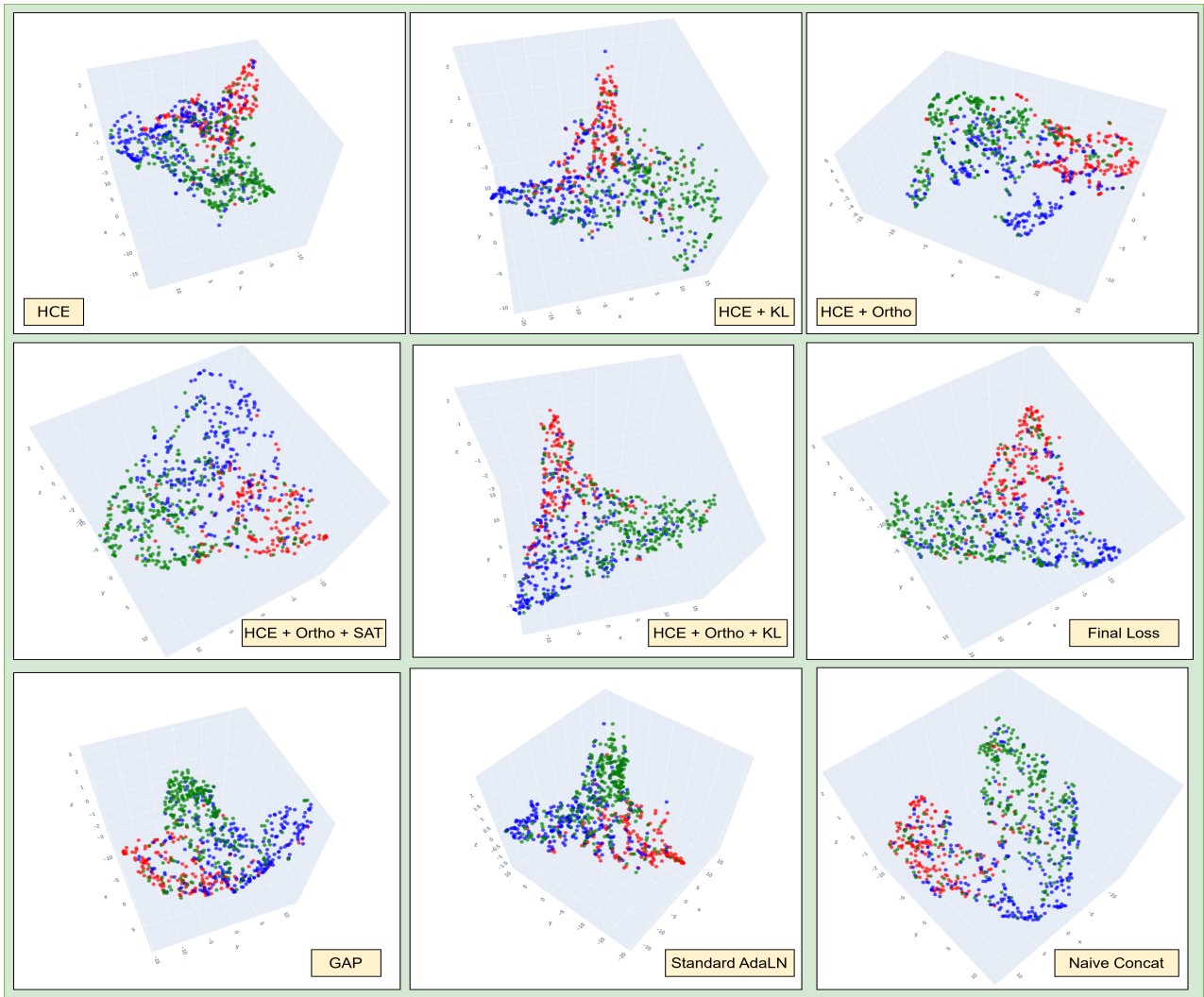

*Figure 4.* t-SNE visualization of the learned latent manifold on the VGAF dataset across different experimental configurations. The data points are color-coded by affective class: Green represents Positive, Red represents Negative, and Blue represents Neutral. We compare the feature space topology of Objective Function Ablations (varying combinations of $\mathcal{L}_{HCE}$ with structural $\mathcal{L}_{ortho}$, $\mathcal{L}_{KL}$ and semantic $\mathcal{L}_{sat}$ regularizers) and Baseline Fusion Strategies (GAP, Standard AdaLN, Naive Concat). The visualization demonstrates that removing components leads to diffuse clusters with significant class overlap. In contrast, the subplot labeled *Final Loss*, which represents the full VIBE framework, exhibits the most distinct class separation and compact clustering, confirming that the synergistic integration of Decoupled AdaLN and the composite loss function effectively disentangles affective signals from noise.

*Table 9.* Pearson correlation analysis between $\mathbf{Z}_{aff}$ and $\mathbf{Z}_{env}$ on the VGAF and GECV validation sets, where Abs. represents the absolute value of the correlation coefficient.

| Metric | VGAF | GECV |
|---|---|---|
| Overall Correlation ($\rho$) | -0.0174 | 0.0054 |
| Mean Abs. Per-Dim Correlation | 0.1045 | 0.1651 |

*Table 10.* Quantitative Disentanglement Validation Metrics for the VGAF dataset, assessing non-linear independence, linear probing accuracy, and causal intervention robustness.

| Metric | Target | Score |
|---|---|---|
| Non-Linear Independence (HSIC) | $\mathbf{Z}_{aff}$ vs $\mathbf{Z}_{env}$ | 0.0015 |
| Linear Probing (Nuisance) | Environment ($\mathbf{Z}_{env}$) | 36.01% |
| Linear Probing (Affective Signal) | Affective ($\mathbf{Z}_{aff}$) | 55.72% |
| Causal Intervention (Latent Swap) | $\Delta y$ upon $\mathbf{Z}_{env}$ swap | 0.2412 |

# E. Global Context Integration and Pooling Strategies

To validate the effectiveness of our proposed Decoupled AdaLN and spatial attention mechanisms, we compare our framework against three alternative strategies for integrating global environmental context with local agent dynamics.

### E.1. Learned Pooling

Unlike standard pooling methods that treat all spatial regions equally, our framework employs a *Learned Query Attention* mechanism. We introduce a learnable query vector, $\mathbf{q}_{pool} \in \mathbb{R}^D$, that serves as a static template for relevant environmental features. Given the global spatial feature map $\mathbf{V}_{global} \in \mathbb{R}^{T \times S \times D}$ (where $S$ is the number of spatial patches), we compute an attention score map $A \in \mathbb{R}^{T \times S}$ via a dot-product with the non-linear projection of the visual tokens:

$$\alpha_{t,i} = \text{Softmax}\left(\mathbf{q}_{pool}^{\top} \cdot \tanh(W_{attn}\mathbf{V}_{global}^{(t,i)})\right) \tag{26}$$

$$\mathbf{c}_{global}^{(t)} = \sum_{i=1}^{S} \alpha_{t,i} \cdot \mathbf{V}_{global}^{(t,i)} \tag{27}$$

The effectiveness of this mechanism stems from its ability to maximize the *Signal-to-Noise Ratio (SNR)* of the context embedding. In unconstrained video scenes, emotional cues are often spatially sparse localized to specific faces, bodies, or interactions amidst large irrelevant background areas. While standard Global Average Pooling (GAP) dilutes these intense but sparse signals by averaging them with low-information background pixels, Learned Query Pooling acts as a *semantic filter*. By learning a query vector $q_{pool}$ that correlates with social activity, the model effectively suppresses background noise and amplifies the relevant sociological context.

### E.2. Global Average Pooling (GAP)

In the GAP variant, we replace this attention mechanism with standard global average pooling. Given the spatial feature map $V_{global}$ , the global context vector $c_{gap}$ is computed as the mean across the spatial dimension:

$$\mathbf{c}_{gap}^{(t)} = \frac{1}{S} \sum_{i=1}^{S} \mathbf{V}_{global}^{(t,i)} \tag{28}$$

This ablation tests the hypothesis that selective attention is necessary to extract meaningful environmental cues, rather than an indiscriminate summary of the entire video frame.

### E.3. Naive Concatenation (Early Fusion)

This variant evaluates the necessity of *feature modulation* versus simple *feature aggregation*. Instead of using our specialized normalization layer to inject context information, the *Naive Concat* approach performs early fusion.

We project the global scene vector to match the dimension of the local agent features and concatenate them along the channel axis. The fused representation $h_{fused}$ is then compressed back to the latent dimension using a linear projection $W_p$:

$$\mathbf{h}_{fused} = W_p \left( \text{Concat}(\mathbf{h}_{local}, \mathbf{h}_{global}) \right) \tag{29}$$

Unlike our proposed method, which modulates the mean and variance of the agent features based on the context, this approach treats the context as just another input signal. This comparison highlights the superiority of treating context as a condition rather than a feature.

### E.4. Standard AdaLN (Static Modulation)

Our proposed *Decoupled AdaLN* is dynamic: it scales the features based on both the static scene ($s$) and the instantaneous interaction intensity ($\gamma$).

The Standard AdaLN variant represents a static physics model. It generates the affine transformation parameters using *only* the global scene token, completely ignoring the synchronization intensity $\gamma$. The modulation is defined as:

$$\text{AdaLN}(\mathbf{h}, \mathbf{s}) = \text{Norm}(\mathbf{h}) \cdot (1 + f_{scale}(\mathbf{s})) + f_{shift}(\mathbf{s}) \tag{30}$$

where $h$ is the input feature vector. By comparing this against our proposed method (where the scale is $f_{scale}(s) + \gamma \cdot f_{sens}(s)$), we isolate the contribution of the *interaction intensity*. This ablation shows that the environment's influence is not constant.

## F. Derivation and Analysis of Synchrony Matrix

In this section, we detail the algorithm for quantifying *Group Cohesion* from raw visual data and analyze the impact of different aggregation strategies. Our approach transforms low-level trajectory data into a high-level interaction descriptor $\gamma$, which serves as the gating signal for the *Gamma Gated Transformer*.

### F.1. Trajectory Extraction and Velocity Normalization

Let $\mathcal{T} = \{b_k^{(t)}\}_{t=1}^T$ denote the sequence of bounding boxes for agent $k$, where $b_k^{(t)} = [x_1, y_1, x_2, y_2]$. We first compute the centroid $c_k^{(t)}$ for each agent at each time step:

$$\mathbf{c}_k^{(t)} = \left( \frac{x_1 + x_2}{2}, \frac{y_1 + y_2}{2} \right) \tag{31}$$

To capture pure motion directionality invariant to speed or camera depth, we compute the normalized velocity vector $\mathbf{v}_k^{(t)}$:

$$\Delta \mathbf{c}_k^{(t)} = \mathbf{c}_k^{(t)} - \mathbf{c}_k^{(t-1)} \tag{32}$$

$$\mathbf{v}_k^{(t)} = \frac{\Delta \mathbf{c}_k^{(t)}}{\|\Delta \mathbf{c}_k^{(t)}\|_2 + \epsilon} \tag{33}$$

where $\epsilon$ is a small constant to prevent division by zero for stationary agents.

### F.2. The Role of Cosine Similarity in Model Stability

We construct a symmetric correlation matrix $S \in \mathbb{R}^{K \times K}$ where each entry $S_{i,j}$ represents the Cosine Similarity of velocity sequences between agents $i$ and $j$:

$$S_{i,j} = \text{CosSim}(\mathbf{v}_i, \mathbf{v}_j) = \frac{1}{T} \sum_{t=1}^T \mathbf{v}_i^{(t)} \cdot \mathbf{v}_j^{(t)} \tag{34}$$

The utilization of Cosine Similarity is critical for the training stability of the Gamma-Gated Transformer for two specific reasons:

**1. Magnitude Invariance (Scale Independence):** In unconstrained video, an agent's pixel velocity depends on its distance from the camera. Euclidean distance or raw dot products would assign higher weights to foreground agents simply because

their motion vectors are larger. Cosine similarity projects all motion onto the unit hypersphere ($||\mathbf{v}|| = 1$), ensuring that the model learns from the *alignment of intent* rather than the *magnitude of pixels*.

**2. Bounded Gating Signal:** Deep networks are sensitive to the scale of input features. Raw velocity dot products are unbounded $[-\infty, \infty]$, which can lead to exploding gradients when used as multiplicative gating factors in AdaLN layers. Cosine similarity strictly bounds the output to $[-1, 1]$:

$$-1 \leq S_{i,j} \leq 1 \tag{35}$$

This provides a mathematically stable control signal, where 1 represents perfect synchronization, $-1$ represents opposition, and 0 represents independence.

### F.3. Dynamic Masking and Scalar Projection

Real-world videos contain varying numbers of agents $k_{real}$, while our model expects a fixed input size $K_{max}$. Simply averaging $S$ would introduce noise from zero-padded agents (padding dilution) and self-loops (diagonal bias).

To derive the robust scalar $\gamma$, we apply a Dynamic Validity Mask $M \in \{0, 1\}^{K \times K}$:

$$M_{i,j} = \mathbb{1}(i \leq k_{real}) \cdot \mathbb{1}(j \leq k_{real}) \cdot \mathbb{1}(i \neq j) \tag{36}$$

The final Synchrony Scalar $\gamma$ is computed as the mean of the valid off-diagonal entries:

$$\gamma = \frac{\sum_{i,j} S_{i,j} \cdot M_{i,j}}{\sum_{i,j} M_{i,j}} \tag{37}$$

This normalization ensures that $\gamma$ serves as a robust, size-invariant measure of group cohesion, effectively filtering out the noise from padding and self-loops. This scalar is subsequently used as the dynamic gating signal for the Gamma-Gated Transformer, allowing the network to modulate its feature-processing sensitivity in response to the instantaneous intensity of the observed social synchronization.

### F.4. Statistical Profile of Group Synchrony

To evaluate the distribution of motion magnitudes and interaction intensities in diverse scenes, we analyzed the statistical profile of the computed group synchrony values ($\gamma$) across the VGAF dataset. Table 11 details the range, mean, median, and standard deviation of these values for both the training and test sets.

*Table 11.* Distribution of group synchrony ($\gamma$) across the VGAF dataset.

| Dataset Split | Range | Mean | Median | Std. Dev. |
|---|---|---|---|---|
| Train | $[-0.81458, 0.99706]$ | 0.11125 | 0.06653 | 0.25119 |
| Test | $[-0.54320, 0.95987]$ | 0.14313 | 0.09064 | 0.26695 |

As indicated by the medians (0.06653 and 0.09064) falling below their respective means, the distribution of synchrony is slightly right-skewed. This statistical profile accurately reflects the "in-the-wild" nature of the VGAF dataset, demonstrating that subtle, independent, or uncoordinated movements naturally occur more frequently than perfectly synchronized collective motion. Despite the prevalence of these small motion magnitudes, our framework successfully leverages this continuous signal to stably gate environmental influence, proving its robustness across the full spectrum of realistic social interactions.

