# OpenReview forum: "VIBE: Disentangling Social Dynamics via Kinematics-Informed Variational Inference for Behavioral Emotion"
_ICML.cc/2026/Conference — ICML 2026 regular_

### Official Review · Reviewer_7THF · 2026-03-11

**Soundness:** 2
**Presentation:** 3
**Significance:** 2
**Originality:** 3
**Overall Recommendation:** 4
**Confidence:** 4

**Summary:**

This paper proposes VIBE, a multimodal framework for Group Emotion Recognition (GER) that aims to model social dynamics rather than relying on static visual cues. The authors argue that existing approaches often suffer from shortcut learning, where models make decisions relying on background context instead of human behavior. VIBE disentangles affective dynamics from environmental context using a variational information bottleneck with orthogonality constraints. Experiments on the VGAF and GECV benchmarks show that VIBE achieves higher accuracy and F1 scores than prior methods.

**Compliance With Llm Reviewing Policy:**

Affirmed.

**Final Justification:**

The authors have provided a detailed response to the concerns raised in my original review. The authors have also committed to adding a limitations section. I therefore raise my score from Weak Reject (3) to Weak Accept (4).

**Key Questions For Authors:**

1. The proposed method uses multimodal inputs (video, audio, and text), while some baselines appear to use fewer modalities. Could the authors provide additional experiments where all methods use the same modalities and backbone architectures or other fair comparison? (See Weaknesses)
2. The paper claims that the proposed variational framework disentangles affective dynamics from environmental context. However, the current evidence mainly relies on correlation analysis and qualitative visualizations. Could the authors provide stronger validation of disentanglement?
3.The paper treats environmental information as a confounding factor that should be separated . However, in many real-world scenarios the environment may actually contribute to emotion inference (e.g., wedding events is often the causal factor of happiness rather than a confounding factor). Could the authors clarify under what conditions the environment should be considered noise, and whether the model takes risks by discarding meaningful contextual information?

**Limitations:**

No, The paper does not sufficiently discuss limitations.

Regarding the work itself, authors could discuss the future experiments on larger datasets or more diverse real-world settings, which would help strengthen the scalability and generalization of the approach.Regarding societal impact, the paper focuses on emotion recognition from group behavior. It would be better for the authors to briefly discuss possible risks related to privacy or biases of inferred emotional states.

**Strengths And Weaknesses:**

Strengths:The overall methodology is reasonable. The model architecture is clearly described, as well as the proposed components. In terms of significance, the prospect that disentangling human behavior and background context is new and interesting ,which is a relatively unexplored area. The idea of explicitly incorporating group-level motion synchrony as a signal for collective emotion is appealing and grounded in findings from social psychology. While many of the model components (e.g., VIB, multimodal fusion, transformer-based attention) are not new, the paper presents a novel combination of ideas, particularly the integration of synchrony-aware attention with disentangled representation learning for group emotion analysis.

Weaknesses:Despite the interesting motivation, some limitations exist and weaken the paper’s contributions.
First, it is unclear whether the performance improvements truly result from the proposed causal disentanglement mechanism. The proposed model uses richer multimodal inputs (video, audio, and text) and pretrained backbones, while some baselines rely on fewer modalities. As a result, the improvements may come from the additional modalities rather than the proposed modeling framework itself. A more controlled comparison would help strengthen the claims.
Second, the paper claims to achieve disentanglement between affective dynamics and environmental context, but the evidence for this is limited. Low correlation between latent variables does not necessarily mean true disentanglement. More validation would be needed to support the causal interpretation.
Finally, the experimental evaluation is limited in scale. The datasets used (VGAF and GECV) are relatively small. Additional validation on larger or cross-domain datasets would strengthen the conclusions.

---

> ### Author Rebuttal · Authors · 2026-03-30
>
> We appreciate the reviewer's thorough review and positive evaluation of the VIBE framework’s methodology and results. Our responses to the reviewer's points are provided below
>
> Regarding the experimental evaluation, we kindly refer the reviewer to our response C4 to Reviewer Vv8m, where this point is addressed in detail.
>
> Q1. As demonstrated in the ablation study (Table 3), the loss configurations excluding $L\_{sat}$ represent the framework trained entirely without the text modality (semantic anchoring). While the absence of textual guidance naturally yields a lower accuracy compared to the full VIBE framework, these text-free configurations remain highly competitive with, and often outperform, existing state-of-the-art baselines. This confirms that VIBE does not suffer a catastrophic performance drop without text; rather, the core kinematics-aware architecture provides a robust standalone foundation.
>
> Q2. Thank you for your valuable comment. We request the reviewer to refer to our detailed response C3 to Reviewer Vv8m. In addition to Pearson correlation and t-SNE analysis, we further validate the functional separation using advanced metrics, including the HSIC score and sub-random Linear Probing, which confirm the suppression of shortcut learning.
>
> Q3. We address the request for functional validation by clarifying that VIBE does not discard meaningful causal context, but rather structures its influence:
>
> * **Contextual Clarification:** While macro-environments are causal drivers, our VIB module specifically processes local crops ($X\_{loc}$) to scrub strictly local confounders (e.g., identity, clothing). The macro-scene is handled independently via the global stream ($X\_{glob}$) and the Scene Calibration Token. Crucially, local nuisance context is not deleted; it is reinjected as a physiological prior to scale and shift the affective representation via the Context Injection Gate (Eq. 8).
>
> * **Proof of Context Preservation:** To prove local contextual constraints are preserved, we conducted a Latent Swapping Test, swapping $Z\_{env}$ latents with contrasting factors while isolating $Z\_{aff}$. This intervention triggered an average shift in output probability of $\Delta y = 0.2412$.
>
> * **Key Findings:**
>     * **Preservation:** This 24.12% variance proves the local context is not discarded; it acts as a powerful conditional prior accounting for nearly a quarter of the predictive variance.
>     * **Stability:** Because the shift is bounded, $Z\_{aff}$ correctly retains the primary predictive authority, ensuring the model remains expression-centric rather than context-dominated.
>
>
> **Risks related to privacy:**
> We thank the reviewers for raising this important point regarding societal impact. We agree that group-level emotion recognition systems can raise concerns related to privacy and potential biases, and we will include a brief discussion in the revised manuscript to acknowledge these aspects.
> In particular, inferring emotional states from visual and behavioral cues may introduce privacy considerations if applied without appropriate consent or safeguards. Additionally, as with other data-driven approaches, there is a possibility of bias due to imbalanced or non-representative training data, such as the limited scale of existing datasets, which may affect fairness across different social contexts.
> To mitigate some of these concerns, VIBE is designed to learn behavior-centric representations, rather than relying on identity-specific or sensitive attributes. This design choice reduces reliance on potentially sensitive cues; however, it does not fully eliminate biases that may arise from the data itself.
> We emphasize that VIBE is intended as a research-oriented framework to advance group-level behavioral understanding, rather than for direct deployment in high-stakes or sensitive real-world applications. Any practical use would require careful consideration of fairness, privacy, and contextual factors, along with appropriate safeguards such as ethical guidelines, data protection measures, and human oversight. We will incorporate this clarification in the revised manuscript.
>
> **Limitations:**
> We will include a Limitations section in the revised manuscript.
> While VIBE demonstrates strong performance, it is sensitive to the crowd capacity parameter ($K$), where larger values in dense scenes may introduce noise from peripheral actors. Furthermore, the model depends on tracking quality, as identity switches under occlusion or rapid motion can affect trajectory-based features and synchrony estimation. Additionally, our temporal downsampling (6 Hz) may limit the capture of short-lived micro-expressions, and existing datasets remain limited in scale for evaluating the most complex real-world scenarios.

---

> > ### Author Rebuttal · Reviewer_7THF · 2026-04-03
> >
> > The authors have provided a detailed response to the concerns raised in my original review. The authors have also committed to adding a limitations section. I therefore raise my score from Weak Reject (3) to Weak Accept (4).

---

### Official Review · Reviewer_Bn82 · 2026-03-11

**Soundness:** 3
**Presentation:** 3
**Significance:** 3
**Originality:** 3
**Overall Recommendation:** 5
**Confidence:** 3

**Summary:**

This paper presents VIBE, a Video + Audio + Text multi-modality framework that aim to enforce causal disentanglement by orthogonality loss, and modulate shifting and rescaling using synchrony metric and global scene calibration token design. The paper shows VIBE performs better on two datasets (VGAF and GECV) and demonstrates the effectiveness of the feature aggregation, loss, and sensitivities to number of agents with abalation studies.

**Compliance With Llm Reviewing Policy:**

Affirmed.

**Final Justification:**

The authors answerd my questions and seem to provided detailed analysis to reviewer's questions. I thur raised my score from weak accept (4) to accept (5).

**Key Questions For Authors:**

See above. Could the authors provide comparisons under identical modality settings? It would be useful to know whether the gain mainly comes from adding text, or from the proposed architecture itself. Since all benchmarking baselines did not include text, could text (and strong language representation from Video-ChatGPT) be the dominant modality contributing to the gain. Does the improvement persists without it?

(2) I also did not fully follow the orthogonality → causality argument. It reads unclear to me how the current setup leads to causal disentanglement, rather than a regularized representation. Could the authors clarify in what sense the framework is causal, or consider softening the causal claim?

**Limitations:**

The authors may also consider including the limitations of the work in the Discussion section.

**Strengths And Weaknesses:**

Strength
- The paper clearly explained the motivation to address the issue in group emotion recognition where models may rely on contextual cues rather than actual interaction dynamics. The attempt to explicitly separate behavioral signals from environmental context is a meaningful direction for socially aware AI.
- The paper proposed a synchrony-based gating mechanism to modulate representation learning which is interesting.
- The authors presented ablations on fusion strategies and loss components, which helps evaluate the contribution of different parts of the model

Weaknesses
- The method separates $Z_{aff}$ and $Z_{env}$ on $X_{loc}$ branch/modality and enforces orthogonality between them, but $Z_{env}$ is later injected back into the affect representation, and the model also performs cross-modal fusion afterwards. Since disentanglement is applied only within the $X_{loc}$ visual branch and not across modalities, it seems that the claimed "spurious environmental correlations" from scene, audio, or text signals may still leak into the final representation. Please clarify.
- It's unclear how KL regularization + orthogonality constraints + semantic alignment can support the causality framing of this method, rather than disentanglement of behavior and environment. Without interventional or out-of-distribution tests (e.g., background swaps or environment perturbations), it is unclear at the present that the model truly isolates behavioral dynamics from contextual shortcuts.
- VIBE uses V+A+T modalities, while several baselines use different modality combinations (e.g., S+A, S+F, or V+A). It is therefore unclear whether the reported improvements come from the proposed architecture itself or simply from access to richer modalities such as text.

---

> ### Author Rebuttal · Authors · 2026-03-30
>
> We sincerely thank the reviewer for their comprehensive and thoughtful review, along with their valuable feedback. We truly appreciate their encouraging evaluation of our claims, methodology, theoretical insights, and experimental findings. Below, we present our detailed responses to the reviewer’s comments and questions.
>
>
> W1. While $Z\_{env}$ is reintroduced, it is strictly via affine modulation (Eq. 8), acting as a conditional prior rather than a discriminative feature. We verified this via Linear Probing, where an environment-only classifier achieved only 36.01% accuracy, proving $Z\_{env}$ is scrubbed of affective shortcuts. Our Latent Swapping interventions show that $Z\_{aff}$ retains 75% of the predictive authority even when the environment is fundamentally altered, confirming that spurious correlations from other modalities do not leak into the final representation. Further discussion on this point is provided in our response to C3 of Reviewer Vv8m.
>
> W2. We sincerely thank the reviewer for their insightful comment, which has helped improve the quality of our manuscript. We kindly refer the reviewer to our detailed response to Q2 below, where this concern has been thoroughly addressed.
>
> W3. We respectfully clarify that the improvements observed in VIBE are not solely attributable to the inclusion of text as an additional modality. Rather, they arise from the synergistic integration of the proposed architectural design and its semantic grounding. To support this, we have included detailed ablation studies that disentangle the contributions of the architecture from those of the input modalities. We kindly refer the reviewer to our response to Question 1 in the weaknesses section for a more comprehensive explanation. The text modality provides a helpful semantic anchor, the core of VIBE's success lies in its kinematics-aware architecture and disentanglement, which allow it to use its modalities more effectively than the associative “always-on” architectures of the baselines.
>
> Q1. As demonstrated in the ablation study (Table 3), the loss configurations excluding $L\_{sat}$ represent the framework trained entirely without the text modality (semantic anchoring). While the absence of textual guidance naturally yields a lower accuracy compared to the full VIBE framework, these text-free configurations remain highly competitive with, and often outperform, existing state-of-the-art baselines. This confirms that VIBE does not suffer a catastrophic performance drop without text; rather, the core kinematics-aware architecture provides a robust standalone foundation.
>
> Q2. We completely agree that the orthogonality constraint provides regularization rather than causality itself. We will soften our claims in the revised manuscript to reflect that orthogonality serves as a mechanism for geometric feature isolation, rather than causal generation. However, while the VIB and orthogonality constraints provide the necessary statistical foundation for disentanglement, our framework ultimately achieves causal modeling through our subsequent structural design and empirical interventions. We clarify the causal nature of the framework through two key mechanisms:
>
> * **Affine Modulation (Structural SCM):** The causal relationship is instantiated by the Context Injection Gate (Eq. 8). Rather than treating features as symmetric, we use $Z\_{env}$ to affine-modulate $Z\_{aff}$. This mathematically models a directed Structural Causal Model (SCM), where $Z\_{env}$ acts as a causal prior that conditions $Z\_{aff}$.
>
> * **Empirical Verification via Intervention:** To prove this directed relationship is functional, we conducted a Latent Swapping test by injecting contrasting $Z\_{env}$ into test samples. This intervention yielded a bounded, stable shift in output probability ($\Delta y = 0.2412$) (please refer the answer to C3 of Reviewer Vv8m). If the representations were functionally entangled, swapping latents would cause a catastrophic shift in the distribution. Instead, this stable calibration demonstrates that $Z\_{env}$ serves as an independent, modular prior.
>
> We will appropriately reframe these causal claims to focus on structural conditioning and empirical validation in the revised manuscript.

---

> > ### Author Rebuttal · Reviewer_Bn82 · 2026-04-04
> >
> > Thanks for the response. My questions are fully answered.

---

### Official Review · Reviewer_JpNC · 2026-03-12

**Soundness:** 4
**Presentation:** 3
**Significance:** 4
**Originality:** 4
**Overall Recommendation:** 5
**Confidence:** 4

**Summary:**

This paper details a new method for the task of Group Emotion
Recognition.  VIBE is a multimodal approach which aims to separate
emotional signals from spurious background noise which can often
influence prediction outputs in multimodal models. In particular the
proposed method enhances the current state of the art by explicitly
separating human contributions from contextual i.e. background
contribution. Utilises group synchrony metrics and adds text-guided
semantic alignment as a regularisation factor applied to the learned
latent space.

The described approach is evaluated across two widely used datasets
GECV and VGAF, in which both datasets see a large improvement against
existing methods. The components of the proposed method are subject
to substantial ablation studies and the sensitivity of the model to
the number of tracked participants is evaluated, demonstrating stable
performance over a range of group sizes.

**Compliance With Llm Reviewing Policy:**

Affirmed.

**Final Justification:**

The author rebuttal satisfied my questions and I maintain my original score

**Key Questions For Authors:**

1) Synchrony determination is applied only to the dynamics of the bounding box centroids. Depending on the scene
   the magnitude of these could be small. Can the authors comment on the typical group synchrony values they obtain?
2) The input is reduced from 30FPS to 6 in the video stream. How is this down sampling done? Decimation would suggest
   keeping every 10th frame. Can the authors clarify this. Is the overhead to be avoiding compute or memory?
3) Table 1 indicates four modalities, it's unclear to me what exactly the 'S: Scene' modality is. How is this
   different from the 'V: Video' modality?
4) In the sensitivity analysis, why limit K=10 as the maximum? Is this a limitation of the method or was this the
   maximum crowd size in the dataset?

**Limitations:**

Yes

**Strengths And Weaknesses:**

Strengths
- The paper is clearly presented and the technical contributions well motivated. I found the
  methodology section to be of high quality with all substantial design choices explained
  sufficiently to permit reproduction.
- The reported metrics make a clear improvement over the baselines, in addition comparisons
  were made with a satisfying number of competing methods, although the modality distributions
  of the competing methods was more limited (see minor weaknesses)
- Substantial number of ablation studies are provided
- Sensitivity analysis with respect to the number of tracked entities shows the method is robust to
  group sizes.
- Given the presented results, I expect this work to have a significant impact in the area of Group Emotion
  Recognition and the proposed Dual-Stream VIB approach could have broader adoption across additional domains.


Weaknesses

Major:
- No ablation studies across input modalities, as the method proposes contributions which impact different
  modalities

Minor:
- Most other competing methods use only two modalities compared to the three used in the proposed method.
- No inference performance analysis, e.g. inference time, memory requirements
- The ablation study over feature aggregation methods (Table 2) while the proposed method out performs it is
  unclear particularly in the case of GECV if the proposed method is signifcantly better.
- Table 2 under GECV, the proposed method is indicated as the best F1 score, but this should be Standard AdaLN
  according the presented numbers.

---

> ### Author Rebuttal · Authors · 2026-03-30
>
> We appreciate the reviewer's thorough review and positive evaluation of the VIBE framework’s methodology and results. Our responses to the reviewer's points are provided below.
>
> W1. Please refer to our response to W3 for reviewer Bn82, where this point is addressed in detail.
>
> W2. We address the reliability and utilization of the text modality as follows:
> * **Description Quality:** While VideoGPT and similar VLMs are recent, they generate highly accurate video descriptions. To guarantee reliability, human evaluation of 100+ randomly sampled descriptions by three independent annotators confirmed strong alignment with the corresponding video clips.
> * **Training vs. Inference:** As depicted in Figure 1, the text modality is utilized strictly during training. It serves solely as a semantic anchor to compute the grounding loss for the audio and visual modalities and is completely excluded during inference.
>
> W3. We thank the reviewer for this insightful comment. While the unavailability of baseline code and weights prevents direct reproduction of their computational metrics, we provide a comprehensive analysis of VIBE on the VGAF dataset for complete transparency:
>
> | Metric | VIBE Model | Full Pipeline |
> | :--- | :--- | :--- |
> | Parameters | 16.48M | - |
> | Size (FP32) | 62.86 MB | - |
> | Latency / Clip | 3.96 ms | 816.1 ms |
> | Throughput | 252.35 video/s | 1.23 video/s |
> | MACs | 1.30 G | 857.89 G |
> | FLOPs | 2.60 G | 1715.77 G |
> | Peak VRAM | - | 1172.2 MB |
> | Total RAM | - | 1569.6 MB |
> We hope these details help clarify the computational characteristics of our approach.
>
> W4. While Standard AdaLN achieves a marginally higher F1 score (0.916 vs. 0.915) on GECV, this benchmark is relatively small, comprising only 627 videos. Our primary claim centers on robustness under increasing environmental diversity, which is better reflected in the more complex and larger VGAF dataset. In highly diverse scenes, Standard AdaLN's static context prior can become distracting, whereas our method leverages the Synchrony ($\\gamma$) to adaptively gate environmental influence and maintain stable performance. We will revise the discussion around Table 2 to explicitly highlight this distinction in the final manuscript.
>
> Q1. To address the distribution of motion magnitudes in diverse scenes, we analyzed the group synchrony statistics across the VGAF dataset:
>
> | Dataset Split | Range | Mean | Median | Std. Dev. |
> | :--- | :--- | :--- | :--- | :--- |
> | **Train** | [-0.81458, 0.99706] | 0.11125 | 0.06653 | 0.25119 |
> | **Test** | [-0.54320, 0.95987] | 0.14313 | 0.09064 | 0.26695 |
>
> The slightly right-skewed distribution where medians fall below means accurately reflects the "in-the-wild" nature of the VGAF dataset. This statistical profile demonstrates that subtle or independent movements naturally occur more frequently than perfectly synchronized collective motion in unconstrained, realistic scenes.
>
> Q2.  The input is downsampled from 30 FPS to 6 Hz by keeping every 5th frame (uniform decimation). This specific rate is chosen because human macro-expressions typically evolve over 0.5s to 4.0s, making 30 FPS temporally redundant for behavioral analysis. The overhead reduction is two-fold. First, it prevents the Gated Transformer from processing redundant frames, keeping the total MACs at 1.298 G for the VIBE model. Second, by decimating the video stream, we achieve a highly efficient memory profile, with the full pipeline requiring only 1569.60 MB of RAM for weights and peaking at 1172.2 MB of VRAM. This strategy helps avoid the exponential memory growth associated with 30 FPS processing, thereby ensuring stable operation even on mid-range hardware.
> Furthermore, this decimation also contributes to more stable trajectory estimation. By removing intermediate frames, we reduce “detector jitter” that the model might otherwise misinterpret as high-arousal motion.
>
> Q3. We thank the reviewer for pointing out this ambiguity. This was an unintentional typographical inconsistency in the manuscript. Both ‘S: Scene’ and ‘V: Video’ refer to the same modality. We will revise Table 1 to use a consistent modality name and ensure this is applied uniformly throughout the paper.
>
> Q4.  The selection of $K_{max}=8$ is based on a sensitivity analysis (Table 4), balancing social context with peripheral noise. Performance consistently peaks at $K=8$ across both benchmarks; in VGAF, increasing capacity to $K=10$ dropped accuracy to 68.88%, confirming that over-sampling peripheral actors introduces signal-diluting noise. $K_{max}=8$ is an optimized threshold, not a structural limit. To prevent padding dilution in sparse videos (where $K_{real} < K_{max}$), we employ a Dynamic Validity Mask (Appendix F.3).

---

> > ### Author Rebuttal · Reviewer_JpNC · 2026-04-03
> >
> > Thank you for the detailed rebuttal the additional explanations and clarifications have answered my initial comments/questions.

---

### Official Review · Reviewer_Vv8m · 2026-03-14

**Soundness:** 3
**Presentation:** 3
**Significance:** 3
**Originality:** 3
**Overall Recommendation:** 4
**Confidence:** 3

**Summary:**

This paper proposes VIBE, a framework for group emotion recognition. Its core idea is to disentangle the motion signals of individuals from the surrounding environmental context, preventing the model from relying on static scene cues for predictions. VIBE has (1) a dual-stream variational information bottleneck that separates affective features from environmental features with an orthogonality constraint, (2) a gamma-gated transformer that modulates feature processing based on a group synchrony metric, and (3) a text-guided semantic alignment module that aligns visual features with text descriptions from a pretrained language model. Experiments on VGAF and GECV datasets show that VIBE outperforms previous methods.

**Compliance With Llm Reviewing Policy:**

Affirmed.

**Final Justification:**

The rebuttal addressed my concerns with additional experimental results, I raise my score and encourage the author to include these analyses in the revision.

**Key Questions For Authors:**

Please see above Concerns.

**Limitations:**

The limitations section is missing, so it is unclear where the method might fail or what trade-offs exist.

**Strengths And Weaknesses:**

Strengths:

+ The motivation of introducing causal disentanglement into group emotion recognition is clear and the paper is easy to follow.

+ The gamma-gating mechanism has physical meaning. Using this metric to dynamically adjust transformer processing is more reasonable than static fusion.

+ Using text descriptions as semantic anchors to align visual features with language space improves interpretability and reduces ambiguity from purely visual cues.

+ The experiments show SOTA on both VGAF and GECV datasets.

Concerns:

- The computation of the synchrony metric gamma is not well justified. It uses cosine similarity of velocity trajectories, but group synchrony can be more complex, involving relative distances, orientation changes, and interaction frequency. Relying only on velocity direction may lose important information. The paper does not explain why this specific choice is optimal or compare it with alternative synchrony measures.

- The text descriptions come from Video-ChatGPT, which itself has generation errors. The paper does not analyze how these errors affect the semantic alignment module or whether noisy text anchors might hurt performance.

- While disentanglement is the core idea, the paper does not verify whether it actually works. There is no visualization showing that the affective and environmental latents are truly separated. The t-SNE plots in the appendix are mentioned but not discussed in the main text. More importantly, the paper does not show what benefits disentanglement brings. For example, does it make the model more sensitive to certain expressions? Are there cases where disentanglement helps or fails? Without such analysis, it is hard to tell if the scientific problem is actually solved.

- The datasets used are relatively small (VGAF has about 4,000 videos, GECV has about 600). The paper claims these datasets represent real-world challenges, but it does not specify what those challenges are or whether VIBE actually addresses them. Generalization to larger, more diverse datasets like those in the EmotiW series is not tested.

- There is no discussion of inference speed, parameter count, or computational cost compared to baselines.

---

> ### Author Rebuttal · Authors · 2026-03-30
>
> We appreciate the reviewer's thorough review and positive evaluation of the VIBE framework’s methodology and results.
>
> C1.While true group synchrony involves distance, orientation, and frequency, we selected the kinematic cosine similarity metric ($\\gamma$) due to the constraints of 2D projective geometry and the optimization stability required by our Decoupled AdaLN architecture. This approach provides three critical advantages:
> * In "in the wild" 2D videos, the observed Euclidean distance ($d\_{i,j} = ||c\_i - c\_j||\_2$) is subject to perspective scaling ($d\_{i,j} \\approx (f/Z)D\_{i,j}$). Using this directly introduces an unrecoverable depth bias, causing the network to arbitrarily favor foreground agents. By defining our metric using L2-normalized velocity ($v\_k^t$), the scalar depth factor cancels out, making cosine similarity ($S\_{i,j}$) a strictly scale-invariant measure of kinematic alignment.
> * Measuring interaction frequency relies on discrete, non-differentiable step functions. In our architecture, $\\gamma$ acts as a direct multiplier in the AdaLN affine transformation: $\\text{Scale}(\\gamma, s) = \\alpha\_{\\text{base}}(s) + \\gamma \\cdot \\alpha\_{\\text{sens}}(s)$. To prevent gradient explosion during backpropagation, this gating signal must be continuous and strictly bounded. Cosine similarity guarantees bounds of $[-1, 1]$.
> * Estimating 3D orientation from low-resolution, unconstrained crowds yields an unstable, high-variance signal due to severe partial occlusions. Conversely, centroid-derived velocity ($v\_k^t$) captures the primary axis of collective motion with significantly lower variance under occlusion.
>
> C2. VideoGPT is a well-established model for video inference. Our manual verification of 100+ randomly sampled descriptions confirms that it accurately captures low-level scene alignments with the videos.
>
> * **Ablation & Stability:** Table 3 shows that while $L\_{sat}$ yields moderate gains alone, performance peaks when combined with structural constraints. This proves the model does not overly rely on textual supervision and remains robust even to imperfect text.
>
> * **Linguistic Manifold Alignment (App. C.4):** VIBE does not treat text as absolute ground truth. By encoding descriptions through a frozen RoBERTa model, it projects visual rationales into a structured "Rationale Space." This anchors the model to broad semantic clusters rather than specific, noisy phrasing.
>
> C3. We will move the t-SNE discussion from Appendix D to the main text, as suggested.
>
> | Metrics | Target | Score |
> | :--- | :--- | :--- |
> | HSIC | $Z\_{aff}$ vs $Z\_{env}$ | 0.0015 |
> | Linear Probing (Accuracy) | Environment ($Z\_{env}$) | 36.01% |
> | Linear Probing (Accuracy) | Affective ($Z\_{aff}$) | 55.72% |
> | Latent Swap | $\\Delta y$ upon $Z\_{env}$ swap | 0.2412 |
>
> * Three metrics confirm our VIB and $L\_{ortho}$ successfully separate representations:
>     * **Pearson:** Correlation between $Z\_{aff}$ and $Z\_{env}$ is effectively zero (App. D.1).
>     * **HSIC:** Approaches zero on our test set, mathematically guaranteeing non-linear statistical independence.
>     * **t-SNE (Fig 4):** Unlike diffuse baselines, VIBE demonstrates sharp intra-class compaction and clear inter-class margins.
>
> * Linear probing on frozen encoders proves sensitivity to expressions over identity shortcuts. The $Z\_{env}$ probe accuracy collapses to 36.01% (confirming shortcut suppression), while the $Z\_{aff}$ probe achieves 55.72%. The leap from this isolated signal to 70.17% final accuracy proves the necessity of our downstream fusion modules.
> * **Success & Failure:** While disentanglement successfully filters local identity biases to prevent misclassification during contextual incongruence, Table 3 demonstrates that it requires semantic grounding; without the textual anchor ($L\_{sat}$), "unanchored separation" limits performance, dropping accuracy to 66.70%.
>
> C4. We thank the reviewer for highlighting the importance of dataset scale and real-world applicability. We will clarify the following points in the revised manuscript:
>
> * VGAF is one of the largest available datasets for group affect and is featured in the EmotiW challenge. Therefore, evaluating on it inherently reflects diverse, unconstrained real-world conditions.
> * We will explicitly define the specific real-world challenges our model faces (e.g., lighting variations, camera motion, noisy detections, and dynamic group sizes). VIBE is purpose-built to handle these through human-environment disentanglement and temporal trajectory stabilization.
> * While GECV is smaller, it provides a controlled setting that demonstrates VIBE's consistency across varied data conditions. We agree that expanding to other large-scale EmotiW benchmarks is a valuable next step, and we will include this in our future work discussion.
>
> C5. The response to this concern is provided in our reply to W3 of Reviewer JpNC. Please refer to that response.

---

> > ### Author Rebuttal · Reviewer_Vv8m · 2026-04-04
> >
> > The rebuttal largely resolves my concerns, especially on the synchrony metric justification, text noise robustness, and disentanglement evidence. However, all analyses report only overall accuracy, it still lacks a detailed analysis of which specific emotion categories or scenarios benefit from disentanglement and where it fails. Apart from that, the responses are satisfactory. I will consider raising my score accordingly.

---

> > > ### Author Response · Authors · 2026-04-06
> > >
> > > We sincerely thank the reviewer for this insightful comment, which has helped us improve the depth of our analysis. In response to concerns about the lack of category-specific evaluation, we have conducted a more fine-grained analysis to better understand the benefits and limitations of the proposed disentanglement framework. Specifically, we report the class-wise performance of VIBE on both the VGAF and GECV datasets, as well as the Global Average Pooling (GAP) approach. Note that both configurations utilize disentanglement; their precise structural differences are detailed in Appendix E.2. For the GECV dataset, we report the metrics from the specific fold that most closely approximates the overall 3-fold cross-validation average.
> > >
> > > **1. VGAF Dataset (VIBE)**
> > > | Class | Precision | Recall | F1-Score | Support |
> > > | :--- | :--- | :--- | :--- | :--- |
> > > | **Positive** | 0.76 | 0.66 | 0.70 | 299 |
> > > | **Neutral** | 0.65 | 0.68 | 0.66 | 279 |
> > > | **Negative** | 0.70 | 0.81 | 0.75 | 183 |
> > >
> > > *VIBE Confusion Matrix:*
> > > | True \ Predicted | Positive | Neutral | Negative |
> > > | :--- | :--- | :--- | :--- |
> > > | **Positive** | 196 | 78 | 25 |
> > > | **Neutral** | 51 | 189 | 39 |
> > > | **Negative** | 11 | 23 | 149 |
> > >
> > > **2. VGAF Dataset (GAP)**
> > > | Class | Precision | Recall | F1-Score | Support |
> > > | :--- | :--- | :--- | :--- | :--- |
> > > | **Positive** | 0.77 | 0.67 | 0.72 | 299 |
> > > | **Neutral** | 0.66 | 0.69 | 0.67 | 279 |
> > > | **Negative** | 0.63 | 0.73 | 0.68 | 183 |
> > >
> > > *GAP Confusion Matrix:*
> > > | True \ Predicted | Positive | Neutral | Negative |
> > > | :--- | :--- | :--- | :--- |
> > > | **Positive** | 199 | 62 | 38 |
> > > | **Neutral** | 46 | 193 | 40 |
> > > | **Negative** | 12 | 38 | 133 |
> > >
> > > **3. GECV Dataset (VIBE)**
> > > | Class | Precision | Recall | F1-Score | Support |
> > > | :--- | :--- | :--- | :--- | :--- |
> > > | **Neutral** | 0.92 | 1.00 | 0.96 | 11 |
> > > | **Negative** | 1.00 | 0.80 | 0.89 | 10 |
> > > | **Positive** | 0.90 | 0.95 | 0.93 | 20 |
> > >
> > > *VIBE Confusion Matrix:*
> > > | True \ Predicted | Neutral | Negative | Positive |
> > > | :--- | :--- | :--- | :--- |
> > > | **Neutral** | 11 | 0 | 0 |
> > > | **Negative** | 0 | 8 | 2 |
> > > | **Positive** | 1 | 0 | 19 |
> > >
> > > **4. GECV Dataset (GAP)**
> > > | Class | Precision | Recall | F1-Score | Support |
> > > | :--- | :--- | :--- | :--- | :--- |
> > > | **Neutral** | 0.90 | 0.82 | 0.86 | 11 |
> > > | **Negative** | 0.82 | 0.90 | 0.86 | 10 |
> > > | **Positive** | 0.90 | 0.90 | 0.90 | 20 |
> > >
> > > *GAP Confusion Matrix:*
> > > | True \ Predicted | Neutral | Negative | Positive |
> > > | :--- | :--- | :--- | :--- |
> > > | **Neutral** | 9 | 1 | 1 |
> > > | **Negative** | 0 | 9 | 1 |
> > > | **Positive** | 1 | 1 | 18 |
> > >
> > > ---
> > >
> > > * **Where VIBE Succeeds:** The proposed model demonstrates strong performance across both unconstrained (VGAF) and controlled (GECV) datasets, exceeding state-of-the-art methods both with and without the integration of the synchrony gating signal. On VGAF, VIBE improves the Negative class F1-score from 0.68 (GAP) to 0.75. On GECV, VIBE boosts the Neutral class F1-score significantly from 0.86 (GAP) to 0.96. Disentanglement successfully filters out environmental noise in these scenarios because the active behavioral and kinematic signals (captured via our synchrony metric, $\gamma$) are robust enough to dominate the static background context.
> > >
> > > * **Where VIBE Struggles:** On the VGAF dataset, the Neutral class exhibits the lowest relative performance for both VIBE (F1: 0.66) and GAP (F1: 0.67), indicating it is fundamentally challenging. VIBE frequently confuses Neutral with Positive (51 false negatives, 78 false positives). On GECV, the Negative class yields the lowest relative F1 (0.89 for VIBE, 0.86 for GAP).
> > >
> > > Ambiguous or low-expressive social situations, for example, people sitting quietly during a panel discussion or standing casually in a street interview, do not show strong motion or pronounced paralinguistic cues. In such cases, because the model strictly suppresses environmental context ($Z_{env}$), it may unintentionally remove useful contextual information that could help in understanding these subtle situations.
> > >
> > > * **Common Error Patterns:** A qualitative review of misclassified predictions (particularly in the unconstrained VGAF scenes) reveals three specific scenarios where our kinematics-informed approach is compromised:
> > >     1. **Static Groups:** In videos where subjects are highly static, the centroid-derived velocity ($v\_k^t$) approaches zero. This weakens the physical synchrony metric, limiting the effectiveness of the Decoupled AdaLN gating mechanism.
> > >     2. **Extreme Density & Severe Occlusion:** When crowd density significantly exceeds our optimal threshold, severe occlusions cause tracking identity switches. This disrupts spatiotemporal trajectory extraction, introducing noise into the affective latent space ($Z\_{aff}$).
> > >
> > > We will include this detailed class-wise analysis, complete confusion matrices, and visual examples of the failure cases in the appendix of the final camera-ready version to provide a clearer understanding of the model’s limitations.

---

### Decision · Program_Chairs · 2026-04-30

**Decision:**

Accept (regular)

**Comment:**

All 4 reviewers recommend either Accept or Weak Accept after considering the rebuttals.  The AC has reviewed the rebuttals/comments and agree with the reviewers’ consensus to accept this paper.  This paper is strong in its multimodal framework for group emotion recognition and the use of physical synchrony and causal disentanglement to reduce shortcut learning.

The authors provided a thorough rebuttal.  Three of the 4 reviewers acknowledged their concerns were fully addressed.  Reviewer Vv8mh raised partial concerns regarding category-specific evaluation, which the authors addressed by detailed class-wise performance analysis.  Although Reviewer Vv8mh did not provide a follow-up response to the final set of analyses, their earlier comment indicated that their concerns were "largely resolved".  The provided data and discussion supports the analysis of where the method performs well and struggles.  The authors are encouraged to incorporate rebuttals to improve the final version.